

# Foliated field theory and string-membrane-net condensation picture of fracton order

**Kevin Slagle**[1,2][⋆]**, David Aasen**[2,3,4]** and Dominic Williamson**[5]

**1** Walter Burke Institute for Theoretical Physics,
California Institute of Technology, Pasadena, California 91125, USA
**2** Institute for Quantum Information and Matter,
California Institute of Technology, Pasadena, California 91125, USA
**3** Kavli Institute of Theoretical Physics, University of California,
Santa Barbara, California 93106, USA
**4** Microsoft Quantum, Microsoft Station Q, University of California,
Santa Barbara, California 93106-6105 USA
**5** Department of Physics, Yale University, New Haven, CT 06511-8499, USA

⋆ kslagle@caltech.edu

## Abstract

Foliated fracton order is a qualitatively new kind of phase of matter. It is similar to topological order, but with the fundamental difference that a layered structure, referred to as a foliation, plays an essential role and determines the mobility restrictions of the topological excitations. In this work, we introduce a new kind of field theory to describe these phases: a foliated field theory. We also introduce a new lattice model and string-membrane-net condensation picture of these phases, which is analogous to the string-net condensation picture of topological order.



# 1 Introduction

Fracton order [1, 2] is a recently theorized and remarkable type of phase of matter which is characterized by topological excitations with various kinds of mobility constraints. Fracton order has garnered much attention from the community recently, likely because the field is motivated from many different directions. Motivational examples include: analytically tractable models of glassy physics and localization (which results from the mobility constraints of the particles) [3–8]; dualities to elasticity theory of two-dimensional crystals [9–13]; quantum information [14–21]; connections to quantum gravity [22] and holography [23]; and classification and characterization of exotic phases of matter [1, 24–32]. Fractons have also been studied from a wide variety of perspectives, including gauging and ungauging [33–39]; generalizations of symmetry protected topological (SPT) order [40–43]; entanglement [44–47]; deconfined criticality [48]; and the search for more experimentally relevant models [49–52].

There are currently three known kinds of (robust or gauged) fracton models, which are summarized in the table below:

|  | $U(1)$ symmetric tensor gauge theory | foliated (type-I) | type-II |
|---|---|---|---|
| example models | scalar charge [53] | X-cube [37], string-membrane-net (Sec. 3) | Haah's code [14], Yoshida's fractal liquids [54] |
| spectrum | gapless | gapped | gapped |
| charge conservation | conserved dipole moment | conserved on stacks of 2D surfaces | conserved on fractal subsets |
| spacetime structure | Einstein manifolds [55] | foliated manifolds | discrete groups? [56] |

The above models are gauge theories, and have been argued [53,57] or proven [14,37,58] to be stable to arbitrary perturbations as non-trivial zero-temperature phases of matter (in the sense of Ref. [59]). Ungauged versions of the above models also exist [10,33–37]; such models require global, subsystem, or fractal symmetries to protect from perturbations that could lead to a trivial (e.g. a direct product state) phase of matter.

The type-II fracton models [14,37,54] remain the most mysterious. They have mostly consisted of exactly-solvable $Z_N$ qudit models with a gapped energy spectrum. Fractons can be created at corners of fractal operators, and the fracton number is conversed (modulo N) on fractal subsets of the system [37]. Most type-II models do not have any mobile excitations, which allows them to be a partially self-correcting quantum memory [15,18,60]. (Some models, e.g. the Sierpinski prism model [54,61], mix characteristics of type-I and type-II models and have both fractal and string operators.) Recently, these models have been generalized to gapless $U(1)$ models with a proposed field theory description [62]. The fractons are even less mobile in these models, which, unlike the gapped type-II models, may allow for a self-correcting quantum memory [63]. The spacetime structure of the type-II fracton models may have recently been generalized beyond flat space in Ref. [56]; however, an explicit example of a generalized type-II model is currently lacking.

The $U(1)$ symmetric tensor gauge theory models [53,57,64–67] generalize $U(1)$ Maxwell gauge theory. This is done by breaking Lorentz-invariance to impose higher moments of charge conservation, such as conservation of dipole or quadropole moments, or by generalizing scalar charges to e.g. vector charges. These models often (but not always [65]) respect spatial rotation symmetry. As a generalization of $U(1)$ gauge theory, symmetric tensor gauge theories are naturally written as field theories, often with an $E^2 + B^2$ kind of Hamiltonian. The stability of these theories to spatial curvature was recently studied in Ref. [55]. The traceless scalar charge theory was found to be the most stable to spatial curvature and maintained gauge invariance on Einstein manifolds. Other theories required Einstein manifolds with constant or no curvature. The loss of gauge invariance on manifolds with forbidden kinds of curvature physically manifests itself as lifting the mobility constraints of the subdimensional particles, making all particles fully mobile on generically curved manifolds.

Foliated (type-I) fracton models [1,3,32,68,69] can be characterized by their subdimensional particle excitations and a foliation structure [24], which is specified by stacks of layers in various directions. Subdimensional excitations are particles that have mobility restrictions when isolated from other particles. There are three kinds of subdimensional particles in foliated models in 3D:

- **planons**, which can only move along the 2D layers;

- **lineons**, which move along the intersection of two layers; and

- **fractons**, which are stuck at the intersection of three layers.

The layers in the foliation structure can be curved, which results in curved lattice models [24, 70].

The type-I models have mostly consisted of exactly-solvable $Z_N$ qudit lattice Hamiltonians. Recently, field theories for the X-cube model and a 2-foliated lineon model have been derived in Ref. [71] and Appendix B of Ref. [27]. However, these field theories can only be applied to flat foliation structures (e.g. cubic lattices with no curvature).

In this work, we introduce a generalized field theory description of foliated fracton phases with curved foliations. The field theory can describe a large class of abelian foliated fracton phases (see Tab. 1 for examples). This is a new kind of field theory, which couples to the foliation structure instead of a Riemann metric. The foliation structure is described by a set of closed one-forms.

The field theory is inspired by a string-membrane-net model[1] of foliated fracton order, which generalizes the X-cube model. The string-membrane-net model is presented in a similar spirit to the Levin and Wen string-net models [72, 73]. Indeed, the ground state wavefunction can be pictured as a large superposition of 1) strings bound to the two-dimensional layers of the foliation and 2) membranes permeating the three-dimensional bulk, where the strings and membranes are subject to various constraints.

One can view the string-membrane-net model and field theory as a 3D toric code (or 3+1D BF theory) that is penetrated by and strongly coupled to multiple stacks of 2D toric code (or 2+1D BF theory) layers (arranged as in Fig. 1, for example). The coupling of the 3D toric code to the 2D layers results in the mobility constraints of the excitations. For example, when a 3D toric code charge passes through a 2D layer, it leaves behind a 2D toric code charge on the 2D layer. As a result, it costs energy each time a 3D toric code charge moves through a layer, which makes it an immobile fracton at low energy (when there are at least three stacks of layers).[2] In the magnetic sector, 2D toric code fluxes are attached to the end-points of 3D toric code flux string excitations. As a result, a pair of 2D fluxes on two intersecting layers behaves as a lineon since the two fluxes must move along the intersection of the layers since they are bound together by a high-energy 3D toric code flux string.

In Sec. 2, we introduce and study the foliated field theory from a purely field theoretic perspective. In Sec. 3, we introduce a string-membrane-net picture of foliated fracton order and a dual coupled-string-net picture. In Appendix A, we generalize the string-membrane-net model to $(Z_M, Z_N)$ qudits on generic lattices (analogous to Refs. [24, 70]). Both sections are self contained.

## 1.1 Notational Conventions

We always work in three spatial dimensions, i.e. 3+1 spacetime dimensions. Greek letters $\mu, \nu, \rho, \sigma = 0, 1, 2, 3$ denote spacetime indices. Repeated spacetime indices are implicitly summed over. The foliation index $k$ is never implicitly summed over; sums over $k$ are always explicitly written. $\delta^\mu_\nu$ denotes a Kronecker delta where $\delta^\mu_\nu = 1$ if $\mu = \nu$, and $\delta^\mu_\nu = 0$ if $\mu \neq \nu$. We index some of the gauge fields using the foliation index $k$ as a superscript: e.g. $A^k_\mu$. When e.g. $k = 2$, this appears as $A^2_\mu$, which should not be confused with the square of $A_\mu$; we have not used integer superscripts to denote powers of gauge fields in this work.

In lattice models, $X$ and $Z$ denote anticommuting Pauli $\sigma^x$ and $\sigma^z$ operators.

---

[1] An equivalent model was independently derived in Section III.B of Ref. [39] by gauging the subsystem symmetries corresponding to stacks of membrane logical operators of 3+1D toric code.

[2] This is depicted in Fig. 3, where the 3D toric code charge is sourced by $j^\mu$ and the 2D toric code charge is sourced by $J^{\mu k}$. We refer to the 2D toric code charge as a dipole because it can decay into a pair of oppositely charged (in a $Z_N$ model) 3D toric code charges on opposite sides of the layer.

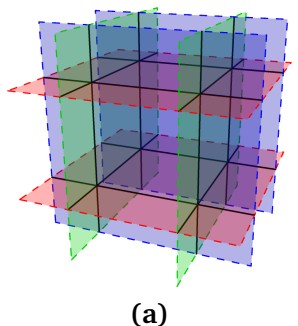
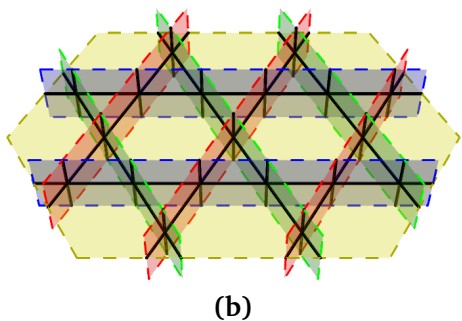

**(a)**          **(b)**

Figure 1: **(a)** A foliation structure consisting of three stacks of layers (red, green, and blue), which are often referred to as leaves. In the coarse-grained continuum limit, we will think of the layers as being infinitesimally close to each other. If the layers have a finite separation, then the intersections of the layers results in a cubic lattice. **(b)** A foliation structure consisting of four stacks of layers (red, green, blue, and yellow), which results in a stack of kagome lattices when the layers have finite separation.

## 2 Foliated Field Theory

Before we can write down a foliated field theory, we must first understand how to describe the foliation structure. Since our focus is on the case of three spatial dimensions, a foliation structure corresponds to a layering structure of one or more stacks of two-dimensional surfaces, exemplified in Fig. 1.

To describe a foliation structure, we introduce a 1-form foliation field $e^k_\mu$ for each $k = 1, 2, .., n_f$. $k$ indexes the different foliations (stack of layers), $\mu$ is a spacetime index, and $n_f$ is the number of foliations. Each $e_\mu$ covector points orthogonal to the foliation layer so that a line integral

$$\int_p e^k \equiv \int_p e^k_\mu \mathrm{d}x^\mu \tag{1}$$

of $e^k_\mu$ over some path $p$ schematically counts the number of layers that the path $p$ crosses; see Fig. 2. However, in the field theory, the layers are spaced infinitesimally close to each other. A cutoff can be added to give a physical meaning to this integral.[3] (On the left-hand-side of the above equality, we are using differential form notation.)

If the path $p$ is open (i.e. is not a closed loop), then the integral $\int_p e^k \in \mathbb{R}$ could be any real number. If the path is closed and contractible, then $\oint_p e^k = 0$ must vanish (Fig. 2). Therefore, $e^k_\mu$ must be a closed 1-form ($de^k = 0$):

$$\partial_\mu e^k_\nu - \partial_\nu e^k_\mu = 0. \tag{2}$$

Physically, this implies that the lattice has no dislocations (where a layer ends), or from another point of view, that spacetime geometry has no torsion [74].

---

[3] We elaborate upon this later in Sec. 4.1.1.

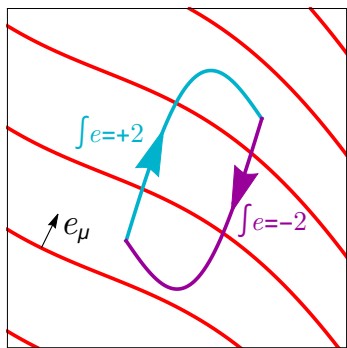

Figure 2: A 2D slice of 3D space showing a single foliation of 2D layers, which appear as red lines in this cross-section. The foliation field $e_\mu$ points orthogonal to the layers such that $\int_p e^k \equiv \int_p e^k_\mu dx^\mu$ counts the number of layers that the path $p$ crosses. In the figure, we show two paths and the resulting integrals. A contractible closed path, e.g. the composition of the cyan and purple paths, results in $\oint_p e^k = 0$. In the field theory, the (red) foliating layers are infinitesimally close to each other. A cutoff can be added to allow for a notion of finite layer spacing.

We can now write down the Lagrangian for a foliated field theory[4]:

$$
\begin{aligned}
L &= \frac{N}{2\pi} \left[ \sum_k e^k \wedge B^k \wedge dA^k + b \wedge da - \sum_k e^k \wedge b \wedge A^k \right] \\
&= \frac{N}{2\pi} \left[ \sum_{k=1}^{n_f} e^k_\mu B^k_\nu \partial_\rho A^k_\sigma + b_{\mu\nu} \partial_\rho a_\sigma - \sum_{k=1}^{n_f} e^k_\mu b_{\nu\rho} A^k_\sigma \right] \epsilon^{\mu\nu\rho\sigma} .
\end{aligned}
\tag{3}
$$

The first line is written in differential form notation, while the second is written with indices. $A^k_\mu$, $B^k_\mu$, and $a_\mu$ are 1-form dynamical gauge fields, and $b_{\mu\nu} = -b_{\nu\mu}$ is a 2-form dynamical gauge field. $\mu, \nu, \rho, \sigma = 0, 1, 2, 3$ are spacetime indices which are implicitly summed over. $k = 1, 2, .., n_f$ indexes the different foliations. After quantization, we expect that $N$ should correspond to the level such that the appropriate lattice model is composed of $Z_N$ qudits.

The first term in the Lagrangian describes a (continuous) stack of 2+1D $Z_N$ gauge theories for each foliation. The second term describes a 3+1D $Z_N$ gauge theory.[5] The third term couples the 2+1D layers to the 3+1D gauge theory.

$e^k_\mu$ is static (i.e. nondynamical) and describes the foliation structure of space. The above field theory is a foliated field theory since it couples to the foliation structure $e^k_\mu$ instead of e.g. a metric $g_{\mu\nu}$, to which most field theories couple. When we want to describe the foliation structure $e^k_\mu$ using a lattice model Hamiltonian, we will require that $e^k_\mu$ has no time component and is constant in time:

$$
e^k_0 = 0, \qquad\qquad \partial_0 e^k_\mu = 0 .
\tag{4}
$$

In Sec. 2.1.1, we show that the number of foliations $n_f$ greatly affects the mobility of the charge excitations that couple to $a_\mu$. Typical examples include are shown in Tab. 1. Lattice models typically consider flat foliations, for which $\partial_\mu e^k_\nu = 0$. When $n_f \leq 3$, it is convenient to choose $e^k_\mu = \delta^k_\mu$, where $\delta^k_\mu$ denotes a Kronecker delta. Fig. 1 (a) and (b) show foliation examples where $n_f = 3$ and $n_f = 4$, respectively.

---

[4] In Sec. 3.3.3, we explain how this field theory was derived.

[5] The first two terms of the Lagrangian utilize 2+1D and 3+1D $Z_N$ BF theory $L = \frac{N}{2\pi} B \wedge dA$, which are continuum descriptions of 2+1D and 3+1D toric code, as reviewed in Appendix A and B of Ref. [71].

Table 1: Charge mobility (to be derived in Sec. 2.1.1) and example models that can be described by different numbers of foliations. It is expected [27] that $n_f = 4$ foliations can describe Chamon's model [3]. (The checkerboard model is equivalent to two copies of the X-cube model [25], and can therefore be described by two copies of the $n_f = 3$ foliated field theory.)

| $n_f$ | 1 | 2 | 3 | 4 |
|---|---|---|---|---|
| charge mobility | planon | lineon | fracton | fracton |
| example lattice models | stack of toric codes | anisotropic lineon model [27] | X-cube [37] | Chamon's model? [3] |

The field theory has a subextensive ground state degeneracy, which we discuss in Appendix B.

## 2.1 Gauge Symmetry and Mobility Constraints

The most interesting aspect of foliated fracton theories is the mobility constraints on their excitations, and that is what we study first. To study excitations, we must first couple the Lagrangian to matter currents ($J^{\mu k}$, $I^{\mu k}$, $j^\mu$, and $i^{\mu\nu}$):

$$L' = L - \sum_k J^{\mu k} A^k_\mu - \sum_k I^{\mu k} B^k_\mu - j^\mu a_\mu - i^{\mu\nu} b_{\mu\nu}. \tag{5}$$

The original Lagrangian [Eq. (3)] is invariant under the following gauge symmetries

$$A^k_\mu \to A^k_\mu + \partial_\mu \zeta^k + \alpha^k e^k_\mu, \qquad\qquad a_\mu \to a_\mu + \partial_\mu \sigma - \sum_k \zeta^k e^k_\mu,$$

$$B^k_\mu \to B^k_\mu + \partial_\mu \chi^k + \lambda_\mu + \beta^k e^k_\mu, \qquad b_{\mu\nu} \to b_{\mu\nu} + \tfrac{1}{2}\partial_\mu \lambda_\nu - \tfrac{1}{2}\partial_\nu \lambda_\mu, \tag{6}$$

where $\chi^k$, $\zeta^k$, $\sigma$, $\lambda_\mu$, $\alpha^k$, and $\beta^k$ are arbitrary functions of the space-time coordinates. It is noteworthy that when $b_{\mu\nu}$ is transformed by $\partial_\mu \lambda_\nu$, $B^k_\mu$ also transforms; similarly, $a_\mu$ also transforms when $A^k_\mu$ is transformed by $\partial_\mu \zeta^k$. This results from the third term in Eq. (3), which strongly couples the $A^k_\mu$ and $b_{\mu\nu}$ fields.[6]

When we require that $L'$ is invariant under these gauge transformations, we have to impose constraints on the currents:

$$\partial_\mu J^{\mu k} \overset{\zeta}{=} -e^k_\mu j^\mu, \qquad \partial_\mu j^\mu \overset{\sigma}{=} 0, \qquad e^k_\mu J^{\mu k} \overset{\alpha}{=} 0, \tag{7}$$

$$\partial_\mu I^{\mu k} \overset{\chi}{=} 0, \qquad \partial_\nu i^{\nu\mu} \overset{\lambda}{=} \sum_k I^{\mu k}, \qquad e^k_\mu I^{\mu k} \overset{\beta}{=} 0. \tag{8}$$

The symbol over each equality sign above denotes which gauge transformation imposes the constraint.

---

[6] As explained around Eq. A12 (and 12) of Ref. [71], the gauge transformations of BF-like field theories can be derived from the equations of motion for the currents [Eq. (11) and (12)]. The mixing of $a_\mu$ and $A^k_\mu$ (and also $b_{\mu\nu}$ and $B^k_\mu$) fields under gauge transformations can then be thought of as resulting from the fact that the currents also mix the gauge fields. In the lattice model [Sec. 3.3], this mixing corresponds to Hamiltonian terms that mix link and plaquette operators.

### 2.1.1 Fracton/Charge Mobility

Let us now consider the constraints imposed on the currents $J^{\mu k}$ and $j^\mu$ in Eq. (7). We refer to $j^\mu$ as a charge current, and $J^{\mu k}$ as a dipole current, for reasons that are explained below. When there are at least three (linearly independent) foliating layers ($n_f \geq 3$), then $j^\mu$ will correspond to a fracton current while $J^{\mu k}$ will describe a planon current.

$\partial_\mu j^\mu = 0$ tells us that the charge current $j^\mu$ must be conserved. However, $\partial_\mu J^{\mu k} = -e^k_\mu j^\mu$ implies that for a given foliation $k$, the divergence of $J^{\mu k}$ must be equal to minus the amount of charge current passing through the foliation $k$. This implies that the $J^{\mu k}$ current can be converted into a dipole of charge current $j^\mu$.

As an explicit example, if a stack of yz-planes is one of the foliations, e.g. if $e^1_\mu = \delta^1_\mu$, then the following currents satisfy the continuity current constraints [Eq. (7)]:

$$
\begin{aligned}
j^\mu &= -\Theta(t)\partial_x \delta^3(\mathbf{x})\delta^\mu_0 + \delta(t)\delta^3(\mathbf{x})\delta^\mu_1\,, \\
J^{\mu 1} &= \Theta(-t)\delta^3(\mathbf{x})\delta^\mu_0\,,
\end{aligned}
\tag{9}
$$

where $\Theta$ is the Heaviside step function, which obeys $\partial_t \Theta(t) = \delta(t)$. The above currents describe a particle of $J^{\mu 1}$ current being transformed (at time $t = 0$) into an $x$-axis dipole of $j^\mu$.[7] Therefore, it makes sense to call $J^{\mu k}$ a dipole current since it can be interchanged for a dipole of charge current $j^\mu$.[8]

$e^k_\mu J^{\mu k} = 0$ implies that for each foliation $k$, the dipole current $J^{\mu k}$ must be orthogonal to $e^k_\mu$, which means that the dipole current is constrained to only move along a layer.

Let us now consider the charge current $j^\mu$ in more detail. Recall that $\partial_\mu J^{\mu k} = -e^k_\mu j^\mu$ implies that for a given foliation $k$, the amount of charge current passing through the foliation $k$ must equal the divergence of the dipole current $J^{\mu k}$. Therefore, in order for charge ($j^\mu$) to pass through a foliation layer $k$, dipoles ($J^{\mu k}$) must be created or absorbed, as depicted in Fig. 3.

If we do not allow the creation of additional particles, this implies that the charge current $j^\mu$ can not pass through foliation layers. If space is foliated by at least three linearly independent foliations, then this implies that the charges are immobile fractons. With only one or two foliations, the charges are planons or lineons, respectively, and as claimed in Tab. 1.

Alternatively, if we start with the vacuum, we can create four charges in the following way. Create a dipole particle (sourced by ($J^{\mu k}$)) and two charges on opposite sides of a layer. The dipole, which is a planon, can then move along the layer and decay into two charges elsewhere. In this way, we see that charges can be created in groups of four, just like the fractons in the X-cube model.

### 2.1.2 Lineon Mobility

Now consider the constraints imposed on the flux currents $I^{\mu k}$ and $i^{\mu\nu}$. We will see that $I^{\mu k}$ describes a conserved flux particle current; but the fluxes will be bound together into lineons, which can only move along the intersection of two foliation layers. $i^{\mu\nu}$ will describe flux string excitations which are less important due to their high energy cost.

$\partial_\mu I^{\mu k} = 0$ implies that the flux $I^{\mu k}$ is conserved, while $e^k_\mu I^{\mu k} = 0$ says that for each foliation $k$, the flux can only move along a layer. However, $\partial_\nu i^{\nu\mu} = \sum_k I^{\mu k}$ tells us that the sum of flux

---

[7] For $t > 0$, the electric dipole moment is given by $\mathbf{p} = \int j^0 \mathbf{x} = \hat{x}$.

[8] We can translate Eq. (9) into the lattice model that we introduce later in Sec. 3. $J^{\mu 1} = \Theta(-t)\delta^3(\mathbf{x})\delta^\mu_0$ describes an excitation of the plaquette term [Eq. (21)] for time $t < 0$. $\delta(t)\delta^3(\mathbf{x})\delta^\mu_1$ says that at time $t = 0$, we act with a $\widetilde{Z}$ operator on the plaquette, which annihilates the plaquette excitation, but creates two cube excitations [Eq. (20)] on the two sides of the plaquette. The two cube excitations that exist for $t > 0$ are described by $-\Theta(t)\partial_x\delta^3(\mathbf{x})\delta^\mu_0$. In the dual coupled-string-net language, this corresponds to taking Fig. 8(c) to Fig. 8(b).

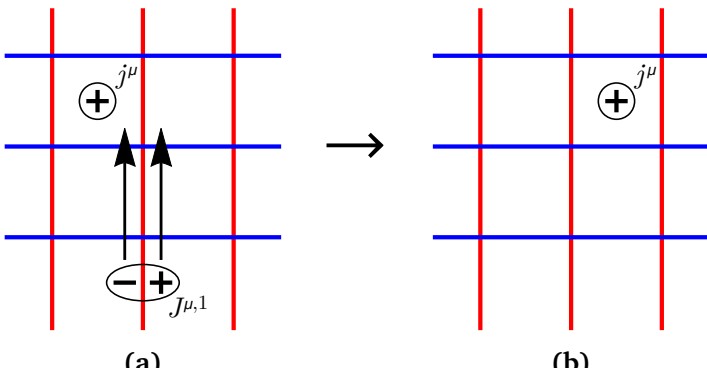

Figure 3: A 2D slice of 3D space showing two foliations (red and blue). **(a)** When in isolation, a charge (sourced by $j^\mu$) cannot move through a foliation layer, which results in the mobility constraints tabulated in Tab. 1. However, a dipole $J^{\mu 1}$ can move along a red layer and be absorbed by the charge to move the charge to the right, as depicted in **(b)**.

currents (for the different foliations) must be equal to the divergence of the string current $i^{\mu\nu}$.[9] However, it costs a lot of energy to make large string excitations. Therefore, we can understand the low-energy mobility restrictions of the particles by only considering small string excitations. If we consider the simplest case of no string excitations, then $i^{\mu\nu} = 0$, which implies that $0 = \partial_\nu i^{\nu\mu} = \sum_k I^{\mu k}$. Therefore, the sum of charge currents for the different foliations must cancel. This implies that the only way a charge on one layer can move is if there is an opposite charge moving along with it on an intersecting layer.

For example, if there are at least two foliations ($n_f \geq 2$) and $e^k_\mu = \delta^k_\mu$ for $k = 1, 2$, then the following currents describe a lineon excitation at the origin:

$$I^{\mu 1} = -I^{\mu 2} = \delta^3(\mathbf{x})\delta^\mu_0. \tag{10}$$

The lineon can't move in the $x$ or $y$ direction because $e^k_\mu I^{\mu k} = 0$ forbids the $I^{\mu 1}$ (or $I^{\mu 2}$) current from moving in the $x$ (or $y$) direction, and $\partial_\nu i^{\nu\mu} = \sum_k I^{\mu k}$ keeps the $I^{\mu 1}$ and $I^{\mu 2}$ currents bound close together (in the absence of high-energy string excitations $i^{\mu\nu}$). In other words, $I^{\mu k}$ describes what would be a planon current (along the yz and zx axis for $k = 1$ and 2), but $\partial_\nu i^{\nu\mu} = \sum_k I^{\mu k}$ confines these planons together into a lineon that can only move in the z-direction.

## 2.2 Equations of Motion

The equations of motion for the foliated field theory [Eq. (3)] coupled to source fields [Eq. (5)] are

$$\epsilon^{\mu\nu\rho\sigma} e^k_\nu (-\partial_\rho B^k_\sigma + b_{\rho\sigma}) \overset{A^k}{\underset{\text{EoM}}{=}} \frac{2\pi}{N} J^{\mu k}, \qquad\qquad \epsilon^{\mu\nu\rho\sigma} \partial_\nu b_{\rho\sigma} \overset{a}{\underset{\text{EoM}}{=}} \frac{2\pi}{N} j^\mu, \tag{11}$$

$$-\epsilon^{\mu\nu\rho\sigma} e^k_\nu \partial_\rho A^k_\sigma \overset{B^k}{\underset{\text{EoM}}{=}} \frac{2\pi}{N} I^{\mu k}, \qquad \epsilon^{\mu\nu\rho\sigma}\left(\partial_\rho a_\sigma - \sum_k e^k_\rho A^k_\sigma\right) \overset{b}{\underset{\text{EoM}}{=}} \frac{2\pi}{N} i^{\mu\nu}. \tag{12}$$

By "$\overset{A^k}{\underset{\text{EoM}}{=}}$", we mean that the equality only holds as an equation of motion for the $A^k$ field, and similar for the other equations.

---

[9] Since $i^{\mu\nu} = -i^{\nu\mu}$ is an antisymmetric tensor with two indices, it describes the current of moving strings. For example, if $I^{\mu k} = 0$, then $i^{10} = -i^{01} = \delta(y)\delta(z)$ (and all other $i^{\mu\nu} = 0$) satisfies $\partial_\nu i^{\nu\mu} = \sum_k I^{\mu k}$ and describes a motionless string excitation along the x-axis.

The equations of motion for $B^k$ and $b$ [Eq. (12)] can be thought of as imposing the string-membrane-net picture [Eq. (18) and (19), or Eq. (15)], which we introduce in the next section. The equations of motion for $A^k$ and $a$ [Eq. (11)] impose the dual string-net picture, which we describe in Sec. 3.4.

# 3 String-Membrane-Net

After reviewing the string-net condensation picture of topological order in Sec. 3.1, we introduce a string-membrane-net picture of foliated fracton order.

## 3.1 String-Net Review

In Ref. [73], Levin and Wen introduced a string-net condensation picture for a wide class of 2+1D topological orders. In this picture, the ground state is given by a weighted superposition over allowed string configurations. In the string-net lattice models, the allowed string configurations and weights are determined by some algebraic data known as a fusion category. Rather than giving a detailed description of this class of models in terms of abstract algebraic data, we demonstrate the construction through a simple example: the toric code, a lattice model for $Z_2$ gauge theory.

The string configurations for toric code are given by coloring the edges of a 2D lattice with $\mathbb{Z}_2$ variables. The coloring is "allowed" if the colored edges form closed loops. The ground state wavefunction is given by an equal-weight superposition of all closed loop configurations:

$$|\Psi\rangle = \cdots + \quad \text{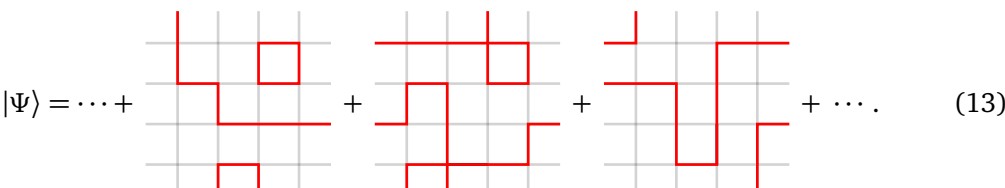} \quad + \quad\quad + \quad\quad + \cdots . \tag{13}$$

The resulting topological order is known as the quantum double of $\mathbb{Z}_2$, sometimes written as $\mathcal{D}(\mathbb{Z}_2)$ [72].

One can specify a Hamiltonian (the toric code [72]) that realizes this wavefunction as its ground state. One can arrive at the Hamiltonian by defining projectors that enforce the two required conditions of the ground state wavefunction: (i) the strings form closed loops, and (ii) all possible closed loops appear with equal weight. To keep track of the string configurations, we place a qubit on every edge and identify the presence of a string with the qubit's eigenvalue under the $Z$ operator (short-hand for $\sigma^z$). If $Z = -1$ on an edge, then we say there is a string on that edge. Condition (i) is enforced by requiring every vertex has an even number of strings entering it, equivalently that $\mathcal{A}_v = \prod_{e \in v} Z_e$ (shown in Fig. 4) has eigenvalue $+1$ on every vertex. Condition (ii) requires all strings fluctuate and "condense"; this is done by adding a term to the Hamiltonian that creates, destroys, and deforms closed strings. Such operations are generated by the plaquette operator $\mathcal{B}_p = \prod_{e \in p} X_e$ (also shown in Fig. 4), and thus the ground state wave function must be a $+1$ eigenstate of $\mathcal{B}_p$ on every plaquette. Hence the Hamiltonian is given by a sum of commuting terms:

$$H = -\sum_p \mathcal{B}_p - \sum_v \mathcal{A}_v, \tag{14}$$

where the first sum is over all plaquettes, and the second is over all vertices. Violations of these terms correspond to local excitations. For example, a charge excitation will have an odd number of strings terminating at a vertex, violating $\mathcal{A}_v$, while a flux excitation corresponds to a violation of $\mathcal{B}_p$.

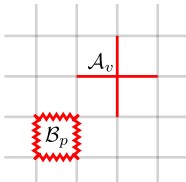

Figure 4: A graphical representation of the two types terms appearing in the toric code Hamiltonian [Eq. (14)]. We use a red cross to represent a product of four $Z$ operators on the edges around a vertex and a red zig-zag to represent a product of four $X$ operators around a plaquette.

## 3.2 String-Membrane-Net

Similar to the toric code example, we begin by introducing the allowed string-membrane-net configurations as an ansatz for the ground state wavefunctions of a fracton model. We then construct a lattice model by writing down a Hilbert space to keep track of these configurations and a Hamiltonian that fluctuates over all allowed configurations. For a certain 3-foliation, the resulting model is equivalent to the X-cube model (up to trivial degrees of freedom and a local unitary), which we show in Sec. 3.3.2.[10]

For expository purposes, we view the cubic lattice as a foliation of either $\mathbb{R}^3$ or $T^3$, depending on context. Following Refs. [24] and [70], we extend this model to any foliated 3-manifold in Appendix A. The leaves (i.e. layers) of the foliation are given by stacks of xy, yz, and zx planes. The edges of the cubic lattice correspond to the intersections of two leaves, while the vertices are given by intersection points of three leaves, as shown in Fig. 1.

A string-membrane-net is given by specifying both a membrane configuration associated with the plaquettes, and a string-net configuration associated with the leaves of the foliation. Here, we focus on the case where the membranes and nets are labeled by $\mathbb{Z}_2$ variables; see Appendix A for the more general construction. Thus, a membrane configuration is specified by an assignment of either $+1$ or $-1$ to each plaquette, indicating the absence or presence of a membrane, respectively. For each leaf, a string configuration is specified by an assignment of $+1$ or $-1$ to each edge in that leaf, which corresponds to the absence or presence of a string on those edges. Since the edges always occur at the intersection of two planes, one must specify a pair of $\mathbb{Z}_2$ values on each edge to specify the entire string-net configuration.

Let $M$ be the set of all possible membrane configurations and $S$ be the set of all possible string configurations residing on the leaves. A string-membrane-net $(m, s) \in M \times S$ is "allowed" if it satisfies:

$$\partial m = \sum_{\ell}^{\text{leaves}} s_{\ell} \qquad \text{and} \qquad \partial s_{\ell} = 0, \qquad (15)$$

where $\ell$ runs over all leaves of the foliation, and $s_{\ell}$ is the string configuration on leaf $\ell$. The first equation says that all edges with an odd[11] number of strings ($s_l$) must be attached to the boundary of a membrane ($\partial m$). The second equation requires that the strings on each leaf

---

[10] In Refs. [69] and [26], p-string condensation and loop condensation pictures of the X-cube model [37] model were presented. Here, we present a similar picture using a string-membrane-net condensation. Our picture has the advantage that it can be realized explicitly as an exactly-solvable lattice model, while the previous condensation pictures were understood perturbatively. We achieve this by introducing quantum degrees of freedom on the faces of the lattice to track the p-strings/loops in the previous condensation pictures.

[11] Here, since there can be at most two strings on an edge, an odd number of strings means exactly one string. In Appendix A, we consider lattice generalizations where multiple leaves can intersect along the same edge so that there can be three or more strings on an edge.

form closed loops (similar to toric code). These constraints are equivalent to the field theory equations of motion in Eq. (12).

In analogy to the toric code, we now stipulate that the ground state wavefunction is given by an equal-weight superposition of all allowed string-membrane-net configurations. Hence, the (un-normalized) ground state wavefunction is given by:

$$|\Psi\rangle = \sum_{m,s}^{\text{allowed}} |(m,s)\rangle. \tag{16}$$

In a very similar way to Eq. (13), we can picture this (unnormalized) wavefunction as

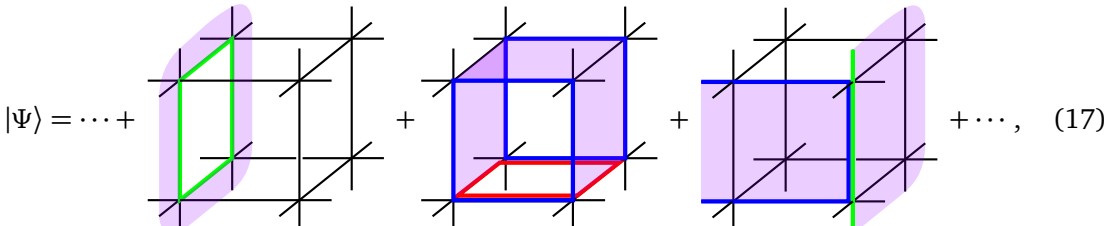

$$|\Psi\rangle = \cdots + \qquad + \qquad + \qquad + \cdots, \tag{17}$$

where the red, green and blue strings belong to the xy, yz, and zx planes, respectively. Edges with a single string always appear at the boundary of a membrane, shown by the shaded purple area. When two different-colored strings overlap on an edge, a membrane does not need to terminate on the edge; one can imagine that there is an infintesimal membrane connecting the two strings.

The excitations in this model correspond to configurations which either do not satisfy the constraint [Eq. (15)] or are not equal-weight superpositions of all possible nets. Before describing the various kinds of excitations, we first define a Hamiltonian.

## 3.3 Lattice Model

We now define a Hamiltonian whose ground state is exactly given by Eq. (17). We first introduce a Hilbert space that allows us to keep track of the string-membrane-nets. We place one qubit on each plaquette $p$ and identify the presence of a membrane with the eigenvalue under $Z_p$. If $Z_p = -1$, then a membrane is present on plaquette $p$. Each edge lives at the intersection of two leaves and therefore requires two qubits to keep track of the string configurations coming from the two leaves. It is convenient to denote the operators acting on this Hilbert space with a superscript that indicates which layer they belong to. For example, an edge $e$ parallel to the x-axis will have two $Z$ operators denoted $Z_e^{zx}$ and $Z_e^{xy}$. If $Z_e^{zx} = -1$ then we say there is a string present the edge $e$ where the string belongs to an zx plane.

The Hamiltonian has four types of terms. The first two enforce the condition that we have an allowed string-membrane-net. The latter two give these string-membrane-nets dynamics and require the ground state is an equal weight superposition over all allowed string-membrane-nets.

Let's first look at the terms that force each layer of the foliation to have a valid net. This is done by the familiar vertex term from the toric code, but applied to every leaf:

$$H_{\text{vert}} = -\sum_v \quad \mathcal{A}_v^{xy} \qquad + \quad \mathcal{A}_v^{yz} \qquad + \quad \mathcal{A}_v^{zx} \tag{18}$$

$$= -\sum_v \left[ Z_{v+\hat{x}}^{xy} Z_{v+\hat{y}}^{xy} Z_{v-\hat{x}}^{xy} Z_{v-\hat{y}}^{xy} + Z_{v+\hat{y}}^{yz} Z_{v+\hat{z}}^{yz} Z_{v-\hat{y}}^{yz} Z_{v-\hat{z}}^{yz} + Z_{v+\hat{x}}^{zx} Z_{v+\hat{z}}^{zx} Z_{v-\hat{x}}^{zx} Z_{v-\hat{z}}^{zx} \right],$$

where a colored edge corresponds to a $Z$ operator acting on that edge in the plane denoted by the superscript of $\mathcal{A}_v$. In the second line, we have written out the operators explicitly. The Hamiltonian is a sum over all vertices ($\sum_v$), and at each vertex we have a cross operator oriented in one of three directions. The cross operator is a product of four $Z$ operators neighboring the vertex. $Z_{v+\hat{x}}^{xy}$ denotes a $Z^{xy}$ operator on the edge in the $+\hat{x}$ direction from the vertex $v$.

We now define the terms enforcing the constraint $\partial m = \sum_\ell s_\ell$. This constraint can be implemented by requiring that the number of membranes whose boundary coincides with a given edge is equal to the number of strings on that edge modulo two. Hence,

$$H_{\text{edge}} = -\overset{\text{x-edge}\ \ \widetilde{\mathcal{A}}_e}{\sum_e} \quad - \overset{\text{y-edge}}{\sum_e}\underset{\widetilde{\mathcal{A}}_e}{} \quad - \overset{\text{z-edge}}{\sum_e}\overset{\widetilde{\mathcal{A}}_e}{} \tag{19}$$

$$= -\overset{\text{x-edge}}{\sum_e} Z_e^{zx} Z_e^{xy} \widetilde{Z}_{e+\hat{y}} \widetilde{Z}_{e+\hat{z}} \widetilde{Z}_{e-\hat{y}} \widetilde{Z}_{e-\hat{z}} \quad - \overset{\text{y-edge}}{\sum_e} Z_e^{xy} Z_e^{yz} \widetilde{Z}_{e+\hat{x}} \widetilde{Z}_{e+\hat{z}} \widetilde{Z}_{e-\hat{x}} \widetilde{Z}_{e-\hat{z}}$$

$$- \overset{\text{z-edge}}{\sum_e} Z_e^{yz} Z_e^{zx} \widetilde{Z}_{e+\hat{x}} \widetilde{Z}_{e+\hat{y}} \widetilde{Z}_{e-\hat{x}} \widetilde{Z}_{e-\hat{y}}.$$

The colored lines denote $Z$ operators acting on the edges from the appropriate leaves, and the purple squares denote Z-operators acting on the plaquettes adjacent to each edge.

We now add terms to the Hamiltonian that force the ground state wavefunction to be an equal weight superposition of the allowed string-membrane-net configurations. We do so by adding terms to the Hamiltonian that fluctuate and condense the allowed string-membrane-nets by creating, destroying, and deforming the strings and membranes. These are generated by two types of terms. The first wraps a membrane over a cube and is given by,

$$H_{\text{vol}} = -\sum_c \quad \boxed{\widetilde{\mathcal{B}}_c} \quad = -\sum_c \prod_{p \in c} \widetilde{X}_p, \tag{20}$$

where the orange sheets represent the action of an $X$ operator on the corresponding plaquette. $\prod_{p \in c}$ is a product over the six plaquettes around the cube $c$. The second type of term lives on the plaquettes in the xy, yz, and zx planes, and is given by,

$$H_{\text{plaq}} = -\overset{\text{xy-plane}}{\sum_p} \boxed{\mathcal{B}_p} \quad - \overset{\text{yz-plane}}{\sum_p} \boxed{\mathcal{B}_p} \quad - \overset{\text{zx-plane}}{\sum_p} \boxed{\mathcal{B}_p} \tag{21}$$

$$= -\overset{\text{xy-plane}}{\sum_p} \widetilde{X}_p \prod_{e \in p} X_e^{xy} - \overset{\text{yz-plane}}{\sum_p} \widetilde{X}_p \prod_{e \in p} X_e^{yz} - \overset{\text{zx-plane}}{\sum_p} \widetilde{X}_p \prod_{e \in p} X_e^{zx}.$$

We have used a notation where a red, green, or blue squiggly line denotes a Pauli $X^{xy}$, $X^{yz}$, or $X^{zx}$ operator on that edge, respectively.

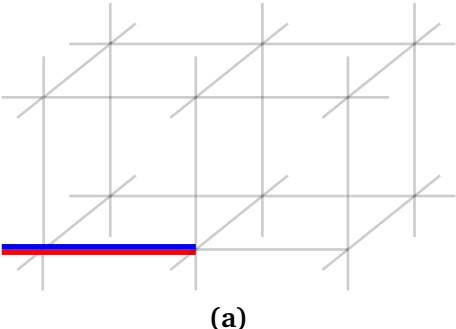
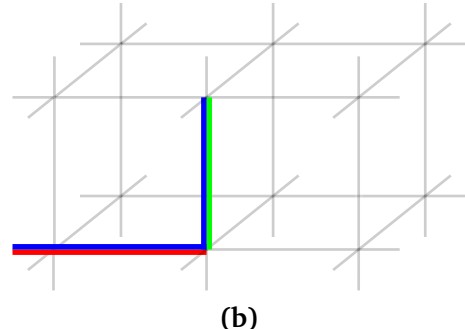

**(a)**                               **(b)**

Figure 5: **(a)** A lineon excitation: two different-colored strings that end at a point. **(b)** The lineon can only move in a straight line since if its path bends, another lineon excitation is left behind at the corner. This occurs because the red string cannot follow the blue string in the $z$-direction as the red string is not allowed on edges in the $z$-direction. The green string is then necessary to avoid excitations of the edge term [Eq. (19)]. But an excitation remains at the corner since the red and green strings both have endpoints there.

Altogether, the Hamiltonian is a sum of four types of terms[12]:

$$H = H_{\text{vert}} + H_{\text{edge}} + H_{\text{plaq}} + H_{\text{vol}}. \tag{22}$$

Next, we describe the excitations found by violating various subsets of these terms.

### 3.3.1 Excitations

In this subsection, we analyze the excitations of the string-membrane-net model.

Let us first consider the string-membrane-net configuration in Fig. 5(a), which shows two overlapping strings along a straight line. The strings end at a point, which violates the closed string constraint [Eq. (18)]. This excitation is a lineon excitation (equivalent to the one in the X-cube model [37]). Lineons can only move along straight lines. If the lineon tries to turn a corner, it will leave behind another lineon excitation at the corner, as shown in Fig. 5(b). If the two different-colored strings try to separate, this will violate the edge term in Eq. (19), which requires that single strings are attached to membranes.

A pair of lineons can form a planon, which can move along a two-dimensional plane. This scenario is depicted in two different ways in Fig. 6.

Excitations of the cube operator [Eq. (20)] correspond to fracton excitations, which are immobile in isolation. In the string-membrane-net picture, fracton excitations correspond to string-membrane-net configurations where negative amplitudes are present in the wavefunction [Eq. (17)]. The fracton excitation is easier to understand in the dual coupled-string-net picture, which we discuss in Sec. 3.4.

A pair of adjacent cube excitations (often called a fracton dipole) is a planon, which can move in the 2D plane straddled by the pair of cubes. This excitation is equivalent to one which only violates the plaquette straddled by the adjacent two cubes [Eq. (21)].

### 3.3.2 Equivalence to X-cube

In this subsection, we show that the low-energy physics of the string-membrane-net model is equivalent to that of the X-cube model [37] by explicitly constructing a local unitary circuit that maps between the two models. In Appendix A, we carry out a similar mapping of the

---

[12] For reference use, we show all of the terms together in Appendix D.

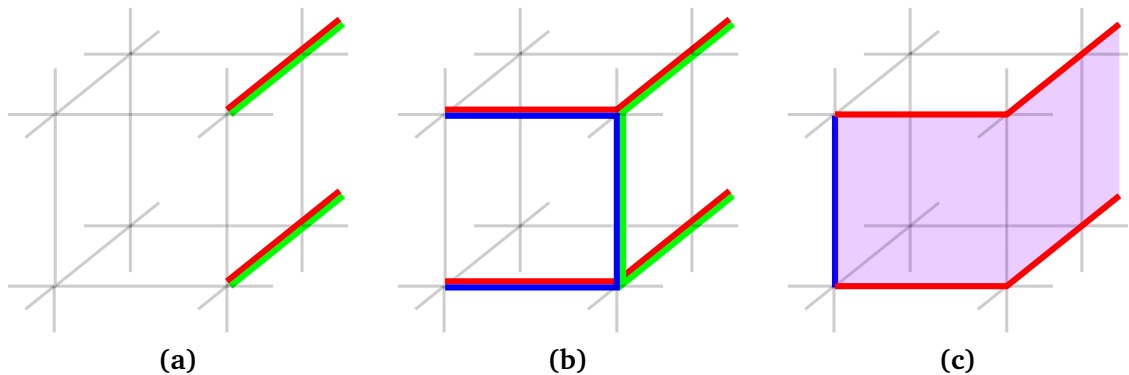

Figure 6: **(a)** A pair of displaced lineons at the two endpoints of the red/green strings. The pair of lineons is a planon, which can move along a two-dimensional plane. This is because, as shown in **(b)**, they can turn a corner (without leaving any excitations behind) by exchanging a blue/green lineon. **(c)** A string-membrane-net configuration that is equivalent to (b) and can be obtained from (b) by applying the string-membrane fluctuation operators in Eq. (21).

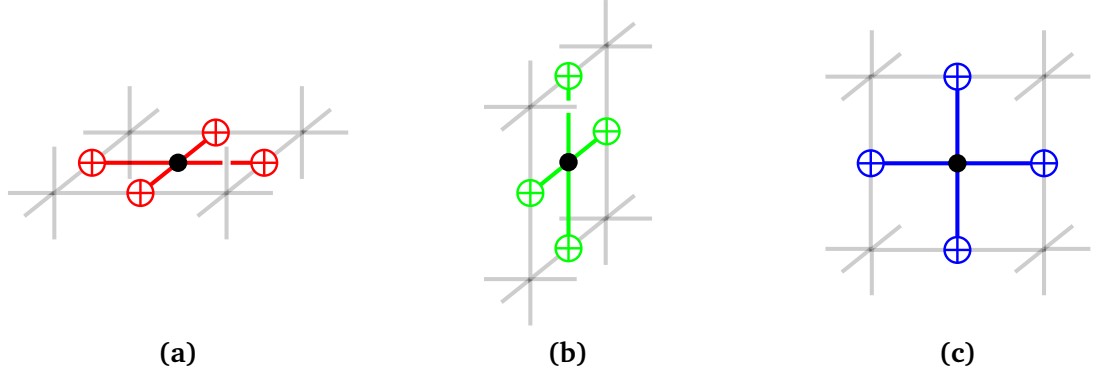

Figure 7: The three components of the unitary transformation given in Eq. (23). The solid dot in the center of each plaquette represents the control qubit for the target qubits on the edges.

once and twice foliated string-membrane-net model on a cubic lattice and show that they are equivalent to a stack of toric codes and the anisotropic lineon model [27], respectively.

Consider the unitary circuit

$$U = \left( \prod_{p}^{\text{xy-plane}} \prod_{e \in p} C_p X_e^{\text{xy}} \right) \left( \prod_{p}^{\text{yz-plane}} \prod_{e \in p} C_p X_e^{\text{yz}} \right) \left( \prod_{p}^{\text{zx-plane}} \prod_{e \in p} C_p X_e^{\text{zx}} \right). \tag{23}$$

The components of this unitary are depicted graphically in Fig. 7. $C_i X_j$ is the controlled-$X$ gate that applies a Pauli $X$ operation to qubit $j$ controlled by the state of qubit $i$, i.e. if $Z_i = -1$. $C_i X_j$ can also be defined by the following commutation relations:

$$(C_i X_j) X_j (C_i X_j)^\dagger = X_j, \qquad\qquad (C_i X_j) X_i (C_i X_j)^\dagger = X_i X_j, \tag{24}$$
$$(C_i X_j) Z_i (C_i X_j)^\dagger = Z_i, \qquad\qquad (C_i X_j) Z_j (C_i X_j)^\dagger = Z_i Z_j.$$

The unitary $U$ acts on the string-membrane-net Hamiltonian [Eq. (18)-(22)] as follows:

$$UH_{\text{vert}}U^\dagger = H_{\text{vert}}, \tag{25}$$

$$UH_{\text{edge}}U^\dagger = -\overset{\text{x-edge}}{\sum_e} Z_e^{\text{zx}}Z_e^{\text{xy}} - \overset{\text{y-edge}}{\sum_e} Z_e^{\text{xy}}Z_e^{\text{yz}} - \overset{\text{z-edge}}{\sum_e} Z_e^{\text{yz}}Z_e^{\text{zx}}, \tag{26}$$

$$UH_{\text{vol}}U^\dagger = -\sum_c \prod_{p\in c} \mathcal{B}_p = -\sum_c \prod_{p\in c}\left(\widetilde{X}_p \prod_{e\in p} X_e^{k(p)}\right), \tag{27}$$

$$UH_{\text{plaq}}U^\dagger = -\overset{\text{xy-plane}}{\sum_p} \widetilde{X}_p - \overset{\text{yz-plane}}{\sum_p} \widetilde{X}_p - \overset{\text{zx-plane}}{\sum_p} \widetilde{X}_p. \tag{28}$$

In Eq. (27), $k(p)$ denotes the xy, yz, or zx plane parallel to the plaquette $p$.

Since Eq. (26) and (28) are sums of terms that each only act locally on a single edge or plaquette, we can view these terms as local constraints that impose

$$Z_e^{\text{zx}}Z_e^{\text{xy}} = 1, \qquad Z_e^{\text{xy}}Z_e^{\text{yz}} = 1, \qquad Z_e^{\text{yz}}Z_e^{\text{zx}} = 1, \qquad \widetilde{X}_p = 1. \tag{29}$$

After imposing these constraints, we are left with a Hilbert space consisting of one effective qubit per edge. The two operators on each edge can be then be mapped to a single operator as follows:

$$Z_e^{\text{xy}} \mapsto Z_e, \qquad Z_e^{\text{zx}} \mapsto Z_e, \qquad X_e^{\text{zx}}X_e^{\text{xy}} \mapsto X_e, \tag{30}$$

for an x-edge $e$, and similar for y and z-edges.

Within this subspace, we recover the X-cube Hamiltonian from Eq. (25) and (27):

$$UHU^\dagger \mapsto H_{\text{X-cube}} \tag{31}$$
$$= -\sum_v \left[ Z_{v+\hat{x}}Z_{v+\hat{y}}Z_{v-\hat{x}}Z_{v-\hat{y}} + Z_{v+\hat{y}}Z_{v+\hat{z}}Z_{v-\hat{y}}Z_{v-\hat{z}} + Z_{v+\hat{x}}Z_{v+\hat{z}}Z_{v-\hat{x}}Z_{v-\hat{z}} \right] - \sum_c \prod_{e\in c} X_e.$$

In Eq. (29), we imposed local constraints on the Hilbert space. This is allowed since we are only trying to show that the string-membrane-net model is in the same phase (as defined in Ref. [59]) as the X-cube model. That is, one can interpolate between the string-membrane-net and X-cube models without passing through a phase transition. If we did not impose the constraints, then we would just be adding trivial gapped degrees of freedom to the X-cube Hamiltonian.

### 3.3.3 Connection to Field Theory

We can make a connection between the lattice model and field theory in the same style as Ref. [71]. See Appendix A of Ref. [71] for the analogous connection between toric code and BF or Chern-Simons theory.

We begin by assuming a rough correspondence between fields and Pauli operators:

$$Z_e^k \sim \exp\left(i\int_{\hat{e}} A^k\right), \qquad X_e^k \sim \exp\left(i\int_e B^k\right),$$
$$\widetilde{Z}_p \sim \exp\left(i\int_{\hat{p}} a\right), \qquad \widetilde{X}_p \sim \exp\left(i\int_p b\right). \tag{32}$$

$Z_e^k$ and $X_e^k$ are the Pauli operators on the edges $e$ of a cubic lattice where $k$ labels the different foliations. In this section, we continue to use $k = \text{yz, zx, xy}$ as an informal version of the

$k = 1, 2, 3$ labelling of the foliations on a cubic lattice. $\widetilde{Z}_p$ and $\widetilde{X}_p$ are the Pauli operators on the plaquettes. The integrals in Eq. (32) denote small integrals over the appropriate edges $e$, dual (on the $k$-plane) edges $\hat{e}$, plaquettes $p$, and dual edges $\hat{p}$ that are dual to the plaquette $p$.

To make a connection to the lattice Hamiltonian, we expand the Lagrangian [Eq. (3)] by separating the time and space parts of the index contractions:

$$
\begin{aligned}
L = & + \frac{N}{2\pi} \underbrace{\Big( \sum_k e_a^k B_b^k \partial_0 A_c^k + b_{ab} \partial_0 a_c \Big) \epsilon^{abc}}_{\text{conjugate fields}} \\
& + \sum_k A_0^k \underbrace{\frac{N}{2\pi} e_a^k \big( -\partial_b B_c^k + b_{bc} \big) \epsilon^{abc}}_{J^{0k}} + a_0 \underbrace{\frac{N}{2\pi} \partial_a b_{bc} \epsilon^{abc}}_{j^0} \\
& + 2 b_{0a} \underbrace{\frac{N}{2\pi} \Big( \partial_b a_c - \sum_k e_b^k A_c^k \Big) \epsilon^{abc}}_{i^{0a}} - \sum_k B_0^k \underbrace{\frac{N}{2\pi} e_a^k \partial_b A_c^k \epsilon^{abc}}_{-I^{0k}} \Big],
\end{aligned}
\tag{33}
$$

where we have made use of the fact that $e^k$ is closed [Eq. (2)] and $e_0^k = \partial_0 e_\mu^k = 0$ from Eq. (4). The $a, b, c = 1, 2, 3$ superscripts and subscripts denote spatial indices (which should not be confused with the $a_\mu$ and $b_{\mu\nu}$ fields).

The first line in Eq. (33) implies that $A$ and $B$ are conjugate fields and that $a$ and $b$ are also conjugate fields. More precisely, if e.g. $e_a^1 = \delta_a^1$ (where $\delta_b^a$ denotes a Kronecker delta), then $B_2^1$ and $A_3^1$ are conjugate fields, and similar for $B_3^1$ and $A_2^1$.[13]

The last four terms are Lagrange multipliers $(A_0^k, a_0, b_{0a}, B_0^k)$ multiplied by expressions that are equal to the equations of motion for the current densities $(J^{0k}, j^0, i^{0a}, I^{0k})$ in Eqs. (11) and (12). When the Lagrange multipliers are integrated out, this results in a constraint that all of these currents are zero. Nonzero currents correspond to excitations. Therefore, the Lagrangian [without coupling to currents in Eq. (5)] describes the ground state Hilbert space with no excitations. Roughly, nonzero currents correspond to excitations of the following operators in the lattice model:

$$
\begin{aligned}
\mathcal{B}^k &\sim \exp\left( i \int J^{0k} \right), & \widetilde{\mathcal{B}} &\sim \exp\left( i \int j^0 \right), \\
\widetilde{\mathcal{A}} &\sim \exp\left( i \int i^{0a} \right), & \mathcal{A}^k &\sim \exp\left( i \int I^{0k} \right).
\end{aligned}
\tag{34}
$$

The integrals above integrate over small spatial regions.

For example, we can view the right-hand-side of the below equation as a continuum version of the $\mathcal{B}$ operator on an zx-plane plaquette when $k = 2$:

$$
\mathcal{B}_p \sim \exp\left[ i \int \underbrace{e_a^k \big( \partial_b B_c^k + b_{bc} \big) \epsilon^{abc}}_{J^{0k} \sim \mathcal{B}^k} \right], \qquad \text{when } k = 2 \text{ and } e_a^2 = \delta_a^2.
\tag{35}
$$

$\mathcal{B}$ is a product of $X^{zx}$ operators on the edges around a zx-plane plaquette and an $\widetilde{X}$ operator at the center of the plaquette. $e_a^k \partial_b B_c^k \epsilon^{abc}$ gives the curl of $B^k$ in the $y$-direction for $k = 2$ (since $e_a^2 = \delta_a^2$), which corresponds to a product of $X^{zx}$ operators around a zx-plaquette on a lattice

---

[13] A simple example of similarly conjugate variables is the Lagrangian for a single Harmonic oscillator where $x$ and $p$ are conjugate variables: $L = p\, \partial_t x - \frac{1}{2} p^2 - \frac{1}{2} x^2$.

[since $X^{zx} \sim \exp\left(i \int B^{k=2}\right)$ in Eq. (32)]. $e_a^k b_{bc}$ corresponds to an $\widetilde{X}$ operator at the center of the plaquette.

We actually originally derived the foliated field theory by making use of the above connection. That is, we first wrote down the string-membrane-net model, and then used relations like Eq. (35) in order to systematically discover the field theory.

### 3.4 Dual Coupled-String-Net Picture

The string-membrane-net picture also has a dual coupled-string-net picture. In this dual picture, we replace the membranes on the direct lattice by strings on the dual lattice. We refer to the strings dual to membranes as "3D strings". Similarly, on each leaf we dualize the strings on the direct square lattice to strings on the dual square lattice. We refer to strings on the 2D leaves as "2D strings". In this dual picture, if $X = -1$ (instead of $Z = -1$) on a edge we say there is a string on that edge. Thus Eqs. (20) and (21) become constraints for the dual coupled-string-net picture. The constraint in Eq. (20) says that the dual 3D strings on the dual cubic lattice must form closed loops. The constraint in Eq. (21) says that the number of dual 2D strings meeting at a vertex from each leaf must equal the number of bulk 3D strings transverse to that vertex modulo two. Eqs. (18) and (19) provide the nets with dynamics and force the ground state to be an equal-weight superposition of all possible nets satisfying the constraints.

In a nutshell, in the dual picture, we have strings on the 2D leaves, and strings describing the 3D toric code; but the 3D toric code strings have to be bound to the endpoint of a 2D string whenever it passes through a layer. This results in a nice picture for the ground state wavefunction:

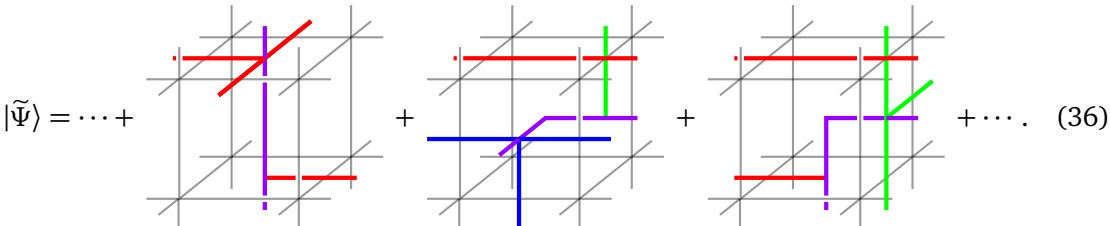

$$|\widetilde{\Psi}\rangle = \cdots + \qquad + \qquad + \qquad + \cdots . \tag{36}$$

The purple 3D strings live on the dual cubic lattice and must always form closed loops. The colored red, green, and blue 2D strings live on the dual square lattice within each leaf. A plaquette can be penetrated by a purple string if and only if a red, green, or blue string ends at the plaquette. This model is equivalent to the string-membrane net, but written in terms of the dual variables.[14]

#### 3.4.1 Fracton Excitation

These dual variables give a nice picture of the fracton excitation, and the fracton dipoles. In Fig. 8(a), we see that fractons are given by the nets that don't satisfy the closed loop condition of the dual 3D toric code strings. In Fig. 8(b), we show a fracton dipole, which is mobile in the plane transverse to the dipole moment. Two of these dipoles can be created locally from the vacuum, which shows that fractons can be created in groups of four, just like in the X-cube model. In Fig. 8(d), we show a gauge-equivalent[15] planon given by a 2D string that is not bound to a 3D string.

---

[14] The coupled-string-net picture can also be viewed as a "p-loop condensate" [69] where the p-loops are given by closed loops of toric code vertex excitations from each layer, rather than plaquette excitations as originally presented in Ref. [69].

[15] In this context, the gauge transformation is generated by the operators that fluctuate the strings: Eqs. (18) and (19) after dualizing the edges and plaquettes, as explained at the beginning of Sec. 3.4.

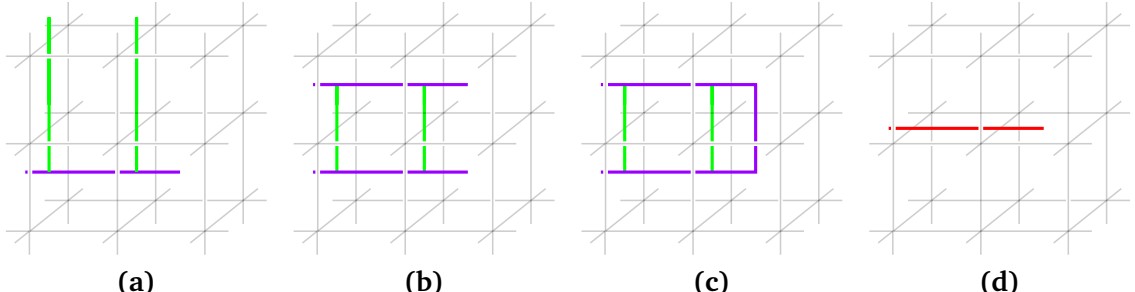

**Figure 8: (a)** An open string (dual to the membranes), which corresponds to a fracton excitation. **(b)** A fracton diople given by two fractons seperated by a plane. It is mobile in the plane orthogonal to the green strings and is therefore a planon. **(c)** A fracton dipole that differs from (b) by only local excitations. **(d)** A planon given by an open string in a 2D plane. This is an equivalent excitation to (c) since both break the constraint that a plaquette can be penetrated by a purple string if and only if a red, green, or blue string ends at the plaquette.

## 4 Conclusion

We have introduced a new foliated field theory and string-membrane-net model of foliated fracton order. The field theory and lattice model (after generalization in Appendix A) both seem to be capable of describing all currently-known abelian foliated fracton orders, such as the ones shown in Tab. 1.

The novel fracton physics of the foliated field theory results from the static foliation space-time structure, which is described by the foliation fields $e^k_\mu$. This is in contrast to most other field theories which couple to a Riemannian metric $g_{\mu\nu}$ (e.g. $U(1)$ Maxwell gauge theory $L = -\frac{1}{4}g^{\mu\rho}g^{\nu\sigma}F_{\mu\nu}F_{\rho\sigma}$ where $F_{\mu\nu} = \partial_\mu A_\nu - \partial_\nu A_\mu$).

It is interesting to note that a foliated field theory can result from a singular limit of the tetradic Palatini field theory of gravity, which we elaborate upon in Appendix C.

A Chern-Simons-like term, which is somewhat similar to the first term of the foliated field theory, also occurs in the topological response of Weyl semimetals [75].

### 4.1 Future Directions

#### 4.1.1 Quantization

One issue that we have left open concerns how to properly quantize the foliated field theory. (The field theory in Ref. [71] also has this issue.) For instance, the field theory naturally describes a continuum of infinitesimally spaced layers along each foliation. But if there is a continuum of layers, it is not clear how to interpret integrals of the foliation field $\int_p e^k$ (Fig. 2). Note that the integral $\int_p e^k$ is dimensionless if we take $e^k$ to have units of inverse length, and it is therefore tempting to interpret noncontractible integrals $\oint_p e^k$ as an integer number of layers. But this does not make sense if there is a continuum of layers. The tendency for a continuum of layers can be seen from Eq. (77), which describes a continuum of degenerate degrees of freedom in the ground state Hilbert space. One could also consider the braiding statistics of the particles (e.g. as in Ref. [71]), and see that there is a continuum of planon particles with nontrivial braiding.

In Appendix B, we show that introducing a cutoff can be used to calculate a finite ground state degeneracy. However, the cutoff methods used in Appendix B was not rigorous. It would be desirable if the ground state degeneracy could be calculated more rigorously.

### 4.1.2  Lattice Model Generalizations

The string-membrane-net picture developed here suggests a generalization by coupling a $(3+1)$D TQFT to layers of $(2+1)$D TQFTs. This could be achieved on the lattice by coupling a generalized Walker-Wang model [76–78] to layers of string-net models. This construction includes models equivalent to the recently introduced cage-net models [32]. The construction can also be viewed as a 3D TQFT with 2D defects, which could provide a possible framework for the future classification of fracton phases. We plan to elaborate on these directions in a forthcoming work.

### 4.1.3  Field Theory Generalizations

One could also imagine generalizing the foliated field theory. For example, we could introduce another 3+1D $Z_N$ gauge theory and couple it to $B^k$ instead of $A^k$:

$$L = \frac{N}{2\pi}\left[\sum_k e^k \wedge B^k \wedge dA^k + b \wedge da + b' \wedge da' - \sum_k e^k \wedge \left(b \wedge A^k + a' \wedge B^k\right)\right]. \tag{37}$$

In the above, $a'$ is a new 2-form gauge field, while $b'$ is a new 1-form gauge field. It is not clear if the above Lagrangian can be described by an exactly-solvable lattice model of qubits using the method in Sec. 3.3.3. One could also consider further generalizing the Lagrangian by adding $M_{IJ}$, $N_I$, and $P_{IJ}$ matrices and vectors as follows:

$$L = \frac{1}{2\pi}\left[\sum_{IJk} M_{IJ}\, e^k \wedge A_I^k \wedge dA_J^k + \sum_I N_I\, b_I \wedge da_I - \sum_{IJk} P_{IJ}\, e^k \wedge b_I \wedge A_J^k\right]. \tag{38}$$

Studying these Lagrangians would be an interesting direction for future work. These Lagrangians may be capable of describing the (abelian) twisted fracton lattice models [31, 41].

### 4.1.4  Dynamical Foliations

The field theory allows us to consider dynamical foliations; i.e. we can consider integrating over all configurations of the foliation field $e_\mu^k$. This can be done by adding an additional term with a new gauge field $f_{\mu\nu}^k$ to the Lagrangian $L$ [Eq. (3)] in order impose the torsion-free constraint [Eq. (2)]:

$$L' = \frac{N}{2\pi}\sum_k \epsilon^{\mu\nu\rho\sigma} f_{\mu\nu}^k \partial_\rho e_\sigma^k. \tag{39}$$

We emphasize that we are now considering both $f_{\mu\nu}^k$ and $e_\mu^k$ as dynamical gauge fields that are integrated over in the path integral. $L + L'$ is not a foliated field theory. Instead, it appears to be a topological quantum field theory (TQFT), similar to the ones studied in e.g. Refs. [79–81]. However, $L + L'$ does not appear to fit into the framework of these works since e.g. the foliation form $e_\mu^k$ does not appear to have a gauge symmetry of the form $e_\mu^k \to e_\mu^k + \partial_\mu \xi^k$, even when the other fields are also allowed to transform under $\xi^k$.

### 4.1.5  More General Foliations

In the math community, it is known that a 1-form foliation field $e$ actually only needs to satisfy

$$de = e \wedge \beta \tag{40}$$

for some 1-form $\beta$. In many simple cases, $\beta$ can be chosen to be zero, which we assumed in Eq. (2). But in some exotic cases, $\beta$ must be nonzero [82, 83]. In fact, the cohomology

class of $\beta \wedge d\beta$ is an invariant of the foliation, which is known as the Godbillon-Vey invariant [84,85]. We leave for future work the generalization of the foliated field theory to foliations with nonzero $\beta$.

## Acknowledgements

We thank Anton Kapustin, Wilbur Shirley, Xie Chen, Zhenghan Wang, Xiao-Gang Wen, Juven Wang, Lei Chen, Alex Turzillo, Meng Cheng, Daniel Bulmash, and Yu An Chen for helpful discussions.

**Funding information**   KS is supported by the Walter Burke Institute for Theoretical Physics at Caltech. DA is supported by a postdoctoral fellowship from the the Gordon and Betty Moore Foundation, under the EPiQS initiative, Grant GBMF4304.

## A   Generalized String-Membrane-Net Model

In this appendix, we extend the string-membrane-net model introduced in Sec. 3 to include more general lattice geometries with $\mathbb{Z}_N$ membranes in the 3D bulk and abelian $\mathbb{Z}_{M_\ell}$ strings on each leaf. The model is defined on a 3D lattice of vertices, edges, and plaquettes, together with a specified set of layers $\ell$ in the lattice.

More formally, the model is defined on any sufficiently-nice[16] cellulation $C$ of 3D space with a specified family of sufficiently-nice[17] cellulated 2D layers $\ell \subset C_2$ embedded in the 2-skeleton $C_2$ of $C$. In many cases of interest, the layers $\ell$ are the leaves of a foliation. We furthermore require that the edges in the 1-skeleton $K_1$ are directed, and that an orientation of the total 3D space, as well as all 2D layers, has been specified.

The Hilbert space is given by a $\mathbb{Z}_N$ qudit on each plaquette and a $\mathbb{Z}_{M_\ell}$ qudit on each edge of each layer (i.e. an edge has a qudit from each layer that contains it):

$$\mathcal{H} = \bigotimes_{p \in C} \mathbb{C}^N \bigotimes_\ell \bigotimes_{e \in \ell} \mathbb{C}^{M_\ell} \,. \tag{41}$$

In the above equation $p$ runs over plaquettes in $C$, $\ell$ runs over layers, and $e$ runs over edges in the $\ell$th layer. Similar to the main text, $\widetilde{Z}_p$ and $\widetilde{X}_p$ Pauli operators act on the plaquettes $p$, and $Z_{e_\ell}$ and $X_{e_\ell}$ act on the edge $e$ from layer $\ell$.[18] The nontrivial commutation relations are

$$\begin{aligned} \widetilde{Z}_p \widetilde{X}_p &= e^{2\pi i/N} \widetilde{X}_p \widetilde{Z}_p \,, \\ Z_{e_\ell} X_{e_\ell} &= e^{2\pi i/M_\ell} X_{e_\ell} Z_{e_\ell} \,. \end{aligned} \tag{42}$$

The Hamiltonian is roughly given by coupling together a 3D $\mathbb{Z}_N$ toric code on the cellulation $C$ with a 2D $\mathbb{Z}_{M_\ell}$ toric code on each layer $\ell$. To define such couplings, we take as input an

---

[16] We require a regular CW complex [86] partitioning space into cells, isomorphic to open balls, such that the boundary of any cell contains a finite number of lower dimensional cells, and any cell only appears in the boundary of finitely many higher dimensional cells.

[17] We assume that each layer is also a CW complex embedded into $C$, which allows the layers to intersect one another, but not themselves. Generalizing to the case of self intersections along edges is straightforward. Unlike Refs. [24,70], we allow more than two layers to intersect along an edge and more than three layers to intersect at a vertex.

[18] Similar to the main text, there can be multiple qudits on an edge, which are distinguished by the layer they belong to. In the main text, we used a superscript to denote which foliation the qudit acts on; in this appendix, we instead use a subscript for the edge label so that we can reserve the superscript for multiplicative powers.

integer $n_\ell$ for each layer such that

$$n_\ell M_\ell = m_\ell N \mod M_\ell N, \tag{43}$$

for some integer $m_\ell$ so that the terms in the resulting model commute with each other.

Let us elaborate on the origin of Eq. (43). We want to allow a subset of the $\mathbb{Z}_N$ membranes to terminate on the $\mathbb{Z}_{M_\ell}$ strings. Let this subset be determined by a map

$$\phi_\ell : \mathbb{Z}_{M_\ell} \to \mathbb{Z}_N, \tag{44}$$

so that if $x \in \mathbb{Z}_{M_\ell}$ labels a string residing in layer $\ell$, then it must live at the boundary of a membrane labeled by $\phi_\ell(x) \in \mathbb{Z}_N$. The map $\phi_\ell$ is not arbitrary, but must be compatible with the fusion rules of the strings and membranes. In particular, the trivial membrane can always terminate on the trivial string, which implies that $\phi_\ell(0) = 0 \mod N$. More generally, we must have $\phi_\ell(a) + \phi_\ell(b) = \phi_\ell(a + b) \mod N$. These two relations tell us that $\phi_\ell$ is a group homomorphism.[19] The kernel of this group homomorphism is composed of the strings in $\mathbb{Z}_M$ that do not need to be attached to a bulk membrane. The image of this group homomorphism is composed of the membranes that are allowed to terminate (on an appropriate string).

Similar to Eq. (22), the Hamiltonian is given by[20]

$$H = H_{\text{vert}} + H_{\text{edge}} + H_{\text{plaq}} + H_{\text{vol}}, \tag{45}$$

with Eq. (18)-(21) generalized as follows:

$$H_{\text{vert}} = -\sum_\ell \sum_{v \in \ell} \prod_{\substack{e \ni v \\ e \in \ell}} Z_{e_\ell}^{\sigma_v^e} + h.c., \tag{46}$$

$$H_{\text{edge}} = -\sum_e \prod_{\ell \ni e} Z_{e_\ell}^{-m_\ell} \prod_{p \ni e} \widetilde{Z}_p^{\sigma_e^p} + h.c., \tag{47}$$

$$H_{\text{plaq}} = -\sum_\ell \sum_{p \in \ell} \widetilde{X}_p^{n_\ell} \prod_{e \in p} X_{e_\ell}^{\sigma_e^p} + h.c., \tag{48}$$

$$H_{\text{vol}} = -\sum_c^{3\text{-cells}} \prod_{p \in c} \widetilde{X}_p^{\sigma_p^c} + h.c. \tag{49}$$

$\sigma_a^b = \pm 1$ is 1 if the orientation on $a$ matches the one induced by $b$. By convention, we take all vertices to be positively oriented, which means that $\sigma_v^e = 1$ if $e$ is directed towards $v$; but this choice does not affect the Hamiltonian. "$h.c.$" denotes the Hermitian conjugate of the preceding terms. $e_\ell$ denotes the qudit on edge $e$ of the layer $\ell$, $p$ denotes a plaquette, and $c$ denotes a 3-cell (i.e. a volume enclosed by plaquettes). $v \in \ell$ denotes a vertex in the layer $\ell$. $\ell \ni e$ denotes a layer that contains the edge $e$. $e \in p$ denotes an edge $e$ at the boundary of the plaquette $p$. $p \ni e$ denotes a plaquette $p$ that has the edge $e$ at its boundary. $e \in \ell$, $p \in \ell$, $p \in c$, $e \ni v$ and $p \in c$ are similar.

---

[19] Specifying a group homomorphism is equivalent to Eq. (43) because, in order to be a homomorphism, $\phi_\ell$ must satisfy $\phi_\ell(M_\ell) = 0$ where $\phi_\ell(x) = n_\ell x \mod N$; this implies that $n_\ell M_\ell = m_\ell N$ for some integer $m_\ell$, which satisfies Eq. (43).

[20] When some of the layers have noncontractible loops with length that does not diverge with system size, the model can have some ground state degeneracy that is not robust to perturbations. This non-robust degeneracy results from the finite-sized (and therefore not robust) logical operators around these finite-sized noncontractible loops. To lift this non-robust degeneracy, additional terms can be added to the model, similar to case for the X-cube model (see e.g. Fig. 4(b-c) of Ref. [70]).

## A.1 Examples

In this subsection, we consider some examples of the string-membrane-net model and show that they map onto previously known models for certain simple foliations. In Sec. 3.3.2, we showed that the string-membrane-net model maps to the X-cube model for $n_f = 3$ orthogonal foliations. More generally, when there are $n_f = 3$ orthogonal foliations with $M_\ell = N$ and $m_\ell = n_\ell = 1$, the model is equivalent to $\mathbb{Z}_N$ X-cube [26]. Another simple example is obtained by setting $m_\ell = n_\ell = 0$, in which case the model reduces to a 3D $\mathbb{Z}_N$ toric code and decoupled layers of 2D $\mathbb{Z}_{M_\ell}$ toric codes.

### A.1.1 Planon model ($n_f = 1$)

In this subsection, we show that a cubic lattice with a single ($n_f = 1$) foliation given by a stack of xy planes with $M_\ell = N = 2$ and $m_\ell = n_\ell = 1$ [defined in Eqs. (42) and (47)-(48)] is equivalent to a stack of decoupled 2D toric codes.

The Hamiltonian for this 1-foliated string-membrane-net model is given by the following terms:

$$H_{\text{vert}}^{(1)} = -\sum_v \; \overset{\mathcal{A}_v^{\text{xy}}}{\diagdown} \; = -\sum_v Z_{v+\hat{x}}^{\text{xy}} Z_{v+\hat{y}}^{\text{xy}} Z_{v-\hat{x}}^{\text{xy}} Z_{v-\hat{y}}^{\text{xy}} , \tag{50}$$

$$H_{\text{edge}}^{(1)} = -\overset{\text{x-edge } \widetilde{\mathcal{A}}_e}{\sum_e} \quad - \overset{\text{y-edge}}{\sum_e} \quad - \overset{\text{z-edge } \widetilde{\mathcal{A}}_e}{\sum_e} \tag{51}$$

$$= -\overset{\text{x-edge}}{\sum_e} Z_e^{\text{xy}} \prod_{p \ni e} \widetilde{Z}_p - \overset{\text{y-edge}}{\sum_e} Z_e^{\text{xy}} \prod_{p \ni e} \widetilde{Z}_p - \overset{\text{z-edge}}{\sum_e} \prod_{p \ni e} \widetilde{Z}_p ,$$

$$H_{\text{plaq}}^{(1)} = -\overset{\text{xy-plane}}{\sum_p} \; \boxed{\mathcal{B}_p} \; = -\overset{\text{xy-plane}}{\sum_p} \widetilde{X}_p \prod_{e \in p} X_e^{\text{xy}} , \tag{52}$$

$$H_{\text{vol}}^{(1)} = -\sum_c \; \boxed{\widetilde{\mathcal{B}}_c} \; = -\sum_c \prod_{p \in c} \widetilde{X}_p . \tag{53}$$

In the above equations, we have used the same graphical notation as in Sec. 3.

In order to map the model to decoupled layers, we consider the following unitary operator

$$U = \left( \overset{\text{xy-plane}}{\prod_p} \prod_{e \in p} C_p X_e^{\text{xy}} \right) \left( \overset{\text{yz-plane}}{\prod_p} \overset{\text{y-edge}}{\prod_{e \in p}} C_p X_e^{\text{xy}} \right) \left( \overset{\text{zx-plane}}{\prod_p} \overset{\text{x-edge}}{\prod_{e \in p}} C_p X_e^{\text{xy}} \right) , \tag{54}$$

where $C_p X_e$ are controlled-$X$ gates, as defined in Eq. (24). The first term is depicted in Fig. 7(a). The second term is a product of controlled-$X^{\text{xy}}$ gates acting on the two neighboring y-axis edges of each yz-plane plaquette. $\prod_{e \in p}^{\text{y-edge}}$ is a product over the y-axis edges $e$ that neighbor the plaquette $p$. The third term is similar.

The above unitary acts on the 1-foliated model as follows:

$$UH_{\text{vert}}^{(1)}U^{\dagger} = -\sum_{v} \quad = -\sum_{v} Z_{v+\hat{x}}^{\text{xy}} Z_{v+\hat{y}}^{\text{xy}} Z_{v-\hat{x}}^{\text{xy}} Z_{v-\hat{y}}^{\text{xy}} \prod_{p \ni v+\hat{z}} \widetilde{Z}_{p} \prod_{p \ni v-\hat{z}} \widetilde{Z}_{p}, \qquad (55)$$

$$UH_{\text{edge}}^{(1)}U^{\dagger} = -\overset{\text{x-edge}}{\sum_{e}} Z_{e}^{\text{xy}} - \overset{\text{y-edge}}{\sum_{e}} Z_{e}^{\text{xy}} - \overset{\text{z-edge}}{\sum_{e}} \prod_{p \ni e} \widetilde{Z}_{p}, \qquad (56)$$

$$UH_{\text{plaq}}^{(1)}U^{\dagger} = -\overset{\text{xy-plane}}{\sum_{p}} \widetilde{X}_{p}, \qquad (57)$$

$$UH_{\text{vol}}^{(1)}U^{\dagger} = -\sum_{c} \prod_{p \in c} \widetilde{X}_{p}. \qquad (58)$$

In Eq. (55), $v+\hat{z}$ denotes the edge in the $+\hat{z}$ direction from the vertex $v$, and $\prod_{p \ni v+\hat{z}}$ denotes the product over all plaquettes $p$ that has the edge $e = v+\hat{z}$ at its boundary.

The above Hamiltonian contains terms that act on single edges and plaquettes. Following Sec. 3.3.2, we view these terms as local constraints that impose

$$Z_{e}^{\text{xy}} = 1, \qquad\qquad \widetilde{X}_{\text{xy-plaquette}} = 1. \qquad (59)$$

This leaves us in a subspace where only the plaquettes in the yz and zx planes are not frozen out.

On this subspace, the Hamiltonian is mapped to

$$UH^{(1)}U^{\dagger} \mapsto -\sum_{v} \quad - \overset{\text{z-edge}}{\sum_{e}} \quad - \sum_{c} \qquad (60)$$

$$= -\sum_{v} \prod_{p \ni v+\hat{z}} \widetilde{Z}_{p} \prod_{p' \ni v-\hat{z}} \widetilde{Z}_{p'} - \overset{\text{z-edge}}{\sum_{e}} \prod_{p \ni e} \widetilde{Z}_{p} - \sum_{c} \overset{\text{yz-plane}}{\prod_{p \in c}} \widetilde{X}_{p} \overset{\text{zx-plane}}{\prod_{p \in c}} \widetilde{X}_{p}.$$

The third term is a sum of products of $\widetilde{X}_{p}$ operators on two yz planes and two zx planes neighboring each cube $c$. This Hamiltonian has the same ground state as a stack of 2D toric code Hamiltonians. The second and third terms behave as 2D toric code cross and plaquette operators (Fig. 4). The first term just changes the energies of the excited states.

### A.1.2  Lineon Model ($n_f = 2$)

In this subsection, we show that a cubic lattice with $n_f = 2$ foliations along the yz and zx planes with $M_{\ell} = N = 2$ and $m_{\ell} = n_{\ell} = 1$ [defined in Eqs. (42) and (47)-(48)] is equivalent to the anisotropic lineon model in Ref. [27].

The Hamiltonian of this 2-foliated string-membrane-net model is given by

$$H_{\text{vert}}^{(2)} = -\sum_v \quad\vcenter{\hbox{$\mathcal{A}_v^{\text{yz}}$}}\quad + \quad\vcenter{\hbox{$\mathcal{A}_v^{\text{zx}}$}} \tag{61}$$

$$= -\sum_v \left[ Z_{v+\hat{y}}^{\text{yz}} Z_{v+\hat{z}}^{\text{yz}} Z_{v-\hat{y}}^{\text{yz}} Z_{v-\hat{z}}^{\text{yz}} + Z_{v+\hat{x}}^{\text{zx}} Z_{v+\hat{z}}^{\text{zx}} Z_{v-\hat{x}}^{\text{zx}} Z_{v-\hat{z}}^{\text{zx}} \right],$$

$$H_{\text{edge}}^{(2)} = -\sum_e^{\text{x-edge } \widetilde{\mathcal{A}}_e} \quad - \sum_e^{\text{y-edge}} \quad - \sum_e^{\text{z-edge } \widetilde{\mathcal{A}}_e} \tag{62}$$

$$= -\sum_e^{\text{x-edge}} Z_e^{\text{zx}} \prod_{p \ni e} \widetilde{Z}_p - \sum_e^{\text{y-edge}} Z_e^{\text{yz}} \prod_{p \ni e} \widetilde{Z}_p - \sum_e^{\text{z-edge}} Z_e^{\text{yz}} Z_e^{\text{zx}} \prod_{p \ni e} \widetilde{Z}_p,$$

$$H_{\text{plaq}}^{(2)} = -\sum_p^{\text{yz-plane}} \vcenter{\hbox{$\mathcal{B}_p$}} \quad - \sum_p^{\text{zx-plane}} \vcenter{\hbox{$\mathcal{B}_p$}} \tag{63}$$

$$= -\sum_p^{\text{yz-plane}} \widetilde{X}_p \prod_{e \in p} X_e^{\text{yz}} - \sum_p^{\text{zx-plane}} \widetilde{X}_p \prod_{e \in p} X_e^{\text{zx}},$$

$$H_{\text{vol}}^{(2)} = -\sum_c \vcenter{\hbox{$\widetilde{\mathcal{B}}_c$}} \quad = -\sum_c \prod_{p \in c} \widetilde{X}_p, \tag{64}$$

where we have used the same graphical notation as in Sec. 3.

In order to map to the lineon model, we consider the following unitary operator

$$U = \left( \prod_p^{\text{xy-plane}} \prod_{e \in p}^{\text{y-edge}} C_p X_e^{\text{yz}} \prod_{e \in p}^{\text{x-edge}} C_p X_e^{\text{zx}} \right) \left( \prod_p^{\text{yz-plane}} \prod_{e \in p} C_p X_e^{\text{yz}} \right) \left( \prod_p^{\text{zx-plane}} \prod_{e \in p} C_p X_e^{\text{zx}} \right), \tag{65}$$

where $C_p X_e$ is defined in Eq. (24). The first term is a product of controlled-$X^{\text{yz}}$ and controlled-$X^{\text{zx}}$ gates acting on the two y-axis and two x-axis edges that neighbor each xy-plane plaquette, respectively. $\prod_{e \in p}^{\text{y-edge}}$ is a product over the y-axis edges $e$ that neighbor the plaquette $p$. The second and third terms are depicted in Fig. 7(b-c).

The above unitary acts on the 2-foliated model as follows:

$$UH_{\text{vert}}^{(2)}U^\dagger = -\sum_v \quad + \quad$$

$$= -\sum_v \Big[ Z_{v+\hat{y}}^{yz} Z_{v+\hat{z}}^{yz} Z_{v-\hat{y}}^{yz} Z_{v-\hat{z}}^{yz} \widetilde{Z}_{v+\hat{x}+\hat{y}} \widetilde{Z}_{v-\hat{x}+\hat{y}} \widetilde{Z}_{v+\hat{x}-\hat{y}} \widetilde{Z}_{v-\hat{x}-\hat{y}} \tag{66}$$

$$+ Z_{v+\hat{x}}^{zx} Z_{v+\hat{z}}^{zx} Z_{v-\hat{x}}^{zx} Z_{v-\hat{z}}^{zx} \widetilde{Z}_{v+\hat{x}+\hat{y}} \widetilde{Z}_{v-\hat{x}+\hat{y}} \widetilde{Z}_{v+\hat{x}-\hat{y}} \widetilde{Z}_{v-\hat{x}-\hat{y}} \Big],$$

$$UH_{\text{edge}}^{(2)}U^\dagger = -\sum_e^{\text{x-edge}} Z_e^{zx} - \sum_e^{\text{y-edge}} Z_e^{yz} - \sum_e^{\text{z-edge}} Z_e^{yz} Z_e^{zx}, \tag{67}$$

$$UH_{\text{plaq}}^{(2)}U^\dagger = -\sum_p^{\text{yz-plane}} \widetilde{X}_p - \sum_p^{\text{zx-plane}} \widetilde{X}_p, \tag{68}$$

$$UH_{\text{vol}}^{(2)}U^\dagger = -\sum_c \quad = -\sum_c \prod_{p\in c} \widetilde{X}_p \prod_{e\in c}^{\text{z-edge}} X_e^{yz} X_e^{zx}. \tag{69}$$

In Eq. (69), $\prod_{e\in c}^{\text{z-edge}}$ is a product over the four z-edges $e$ around the cube $c$.

Again we follow Sec. 3.3.2 and treat the terms acting on a single edge or plaquette as local constraints:

$$Z_{\text{x-edge}}^{zx} = 1, \qquad Z_{\text{y-edge}}^{yz} = 1, \qquad Z_{\text{z-edge}}^{yz} Z_{\text{z-edge}}^{zx} = 1, \qquad \widetilde{X}_p^{yz} = 1, \qquad \widetilde{X}_p^{zx} = 1. \tag{70}$$

This freezes out the x and y edges, yz and zx plaquettes, and leaves one qubit for each xy plaquette and z edge, which we identify via the following mapping:

$$Z_{\text{z-edge}}^{yz} \mapsto Z_{\text{z-edge}}, \qquad Z_{\text{z-edge}}^{zx} \mapsto Z_{\text{z-edge}}, \qquad X_{\text{z-edge}}^{zx} X_{\text{z-edge}}^{xy} \mapsto X_{\text{z-edge}}. \tag{71}$$

Within the subspace satisfying these constraints, the Hamiltonian is mapped to

$$UH^{(2)}U^\dagger \mapsto -2\sum_v \quad - \sum_c \tag{72}$$

$$= -2\sum_v Z_{v+\hat{z}} Z_{v-\hat{z}} \widetilde{Z}_{v+\hat{x}+\hat{y}} \widetilde{Z}_{v-\hat{x}+\hat{y}} \widetilde{Z}_{v+\hat{x}-\hat{y}} \widetilde{Z}_{v-\hat{x}-\hat{y}} - \sum_c \prod_{p\in c}^{\text{xy-plane}} \widetilde{X}_p \prod_{e\in c}^{\text{z-edge}} X_e,$$

where $\prod_{p\in c}^{\text{xy-plane}}$ denotes a product over the two xy-plane plaquettes on the boundary of the cube $c$. There is only one flavor of qubit per z-edge, so the color of the z-edges in the graphical notation is not important. This Hamiltonian is equivalent to the anisotropic lineon model introduced in Ref. [27].

# B   Ground State Degeneracy

Similar to Ref. [71], a finite and subextensive ground state degeneracy of the foliated field theory can be calculated by adding a cutoff to describe the spacing between the foliating layers.

Let us consider a 3-torus with coordinates $0 \leq x < l_x$, $0 \leq y < l_y$, and $0 \leq z < l_z$ and periodic boundary conditions. We shall consider a flat 3-foliation described by

$$e^k_\mu = \frac{L_k}{l_k} \delta^k_\mu, \tag{73}$$

for $k = 1, 2, 3$ where $L_k$ is an integer. This choice of foliation corresponds to a continuum version of the X-cube model on an $L_x \times L_y \times L_z$ cubic lattice. On a periodic $L_x \times L_y \times L_z$ lattice, the $Z_N$ X-cube model has degeneracy [46, 71]

$$\text{GSD} = N^{2L_x + 2L_y + 2L_z - 3}. \tag{74}$$

We will now attempt to reproduce this expression from the field theory.

Fist, we must solve the equations of motion [Eq. (11) and Eq. (12)]. There are many different gauge choices; one choice is the following:

$$A^k_a = \delta(x^a) q^k_a(t, x^k) - \delta(x^a) \delta(x^k) \frac{1 + \epsilon^{kab}}{2} \int_{y^k} q^k_a(t, y^k) + (a \leftrightarrow k),$$
$$B^k_a = \epsilon^{kab} \delta(x^a) p^k_b(t, x^k),$$
$$a_a = 0, \tag{75}$$
$$b_{ab} = 0,$$

$$q^k_a(t, 0) = p^k_a(t, 0) = 0 \text{ for each } k = 1, 2, 3 \text{ and with } a = \begin{cases} 1 & k = 2 \\ 2 & k = 3 \\ 3 & k = 1 \end{cases}. \tag{76}$$

The solution is parameterized by functions $q^k_a(t, x^k)$ and $p^k_a(t, x^k)$. We remark that $q^k_a$ only depends on two coordinates: time $t$ and the spatial coordinate $x^k$. $q$ and $p$ are effectively nonlocal fields that describe the ground state Hilbert space. They require a spatial coordinate for parameterization because the degeneracy of the X-cube model increases with system size. The constraint in Eq. (76) avoids a redundancy and is necessary to reproduce the $-3$ in the degeneracy equation [Eq. (74)].

We can now plug the above solution [Eq. (75)] into the action $S = \int L$ [Eq. (3)]. The result is

$$S = \frac{N}{2\pi} \sum_{k \neq b} \frac{L_k}{l_k} \int_0^{l_k} dx^k \, p^k_b(t, x^k) \partial_t q^k_b(t, x^k), \tag{77}$$

where $\sum_{k \neq b}$ sums over all 6 different choices of $k, b = 1, 2, 3$ such that $k \neq b$. If we ignore quantization issues for the moment, then the above action describes the degenerate Hilbert space of a degree of freedom for each $k \neq b$ and $x^k$, which would give an infinite amount of ground state degeneracy.

In order to obtain a finite ground state degeneracy, one could consider imposing cutoff lengths $a_k \sim \frac{l_k}{L_k}$ in the $x^k$-direction. $x^k$ can then effectively take $\ell_k / a_k = L_k$ different values, and Eq. (77) roughly becomes

$$S \sim \frac{N}{2\pi} \sum_{k \neq b} \sum_{x^k = 0, a_k, \ldots, (L_k - 1)a_k} p^k_b(t, x^k) \partial_t q^k_b(t, x^k). \tag{78}$$

Eq. (78) effectively describes $2L_x + 2L_y + 2L_z - 3$ many $Z_N$ qudits [where the $-3$ comes from Eq. (76)], which matches the ground state degeneracy in Eq. (74). However, a more rigorous derivation of a finite degeneracy (if possible) is left for future work.

# C  Foliated Field Theory from Singular Tetradic Palatini Gravity

In this appendix, we note the interesting curiosity that the Tetradic Palatini action for gravity results in a foliated field theory when linearized about a singuar field configuration.

The Tetradic Palatini action [87] is an alternative to the Einstein-Hilbert action of gravity which has an advantage that it can be written nicely using differential forms. The Lagrangian is

$$L = \frac{1}{16\pi G} \int \epsilon_{\alpha\beta\gamma\delta} \, e^\alpha \wedge e^\beta \wedge \Omega^{\gamma\delta} = \frac{1}{16\pi G} \int \epsilon^{\mu\nu\rho\sigma} \epsilon_{\alpha\beta\gamma\delta} \, e_\mu^\alpha e_\nu^\beta \Omega_{\rho\sigma}^{\gamma\delta} \, \mathrm{d}^4 x, \qquad (79)$$
$$\Omega_{\rho\sigma}^{\gamma\delta} = \partial_\rho \omega_\sigma^{\gamma\delta} + \omega_\rho^{\gamma\alpha} \eta_{\alpha\beta} \omega_\sigma^{\beta\delta} - (\rho \leftrightarrow \sigma),$$

where $\mu, \nu, \rho, \sigma = 0, 1, 2, 3$ are spacetime indices and $\alpha, \beta, \gamma, \delta = 0, 1, 2, 3$ are internal indices, which are both implicitly summed over. $e_\mu^\alpha$ is called a frame-field and factorizes the usual Riemannian metric tensor as $g_{\mu\nu} = e_\mu^\alpha \eta_{\alpha\beta} e_\nu^\beta$ where $\eta_{\alpha\beta}$ is the Minkowski metric. $\Omega_{\rho\sigma}^{\gamma\delta}$ is the curvature of the non-abelian SO(3,1) gauge field $\omega_\mu^{\alpha\beta}$, which $\omega_\mu^{\alpha\beta}$ is called the spin connection. (The frame field is antisymmetric in its upper indices: $\omega_\mu^{\alpha\beta} = -\omega_\mu^{\beta\alpha}$.) The spin connection is related to the usual Christoffel symbols $\Gamma_{\sigma\mu}^\nu$ as $\omega_\mu^{\alpha\beta} \eta_{\beta\gamma} = e_\nu^\alpha \Gamma_{\sigma\mu}^\nu E_\gamma^\sigma + e_\nu^\alpha \partial_\mu E_\gamma^\nu$, where $E_\alpha^\mu$ is the inverse matrix of $e_\mu^\alpha$: $E_\alpha^\mu e_\mu^\beta = \delta_\alpha^\beta$ and $E_\alpha^\mu e_\nu^\alpha = \delta_\nu^\mu$.

At each point in space, the frame field $e_\mu^\alpha$ can the thought of as a matrix (since it has two indices). We can now imagine naively expanding $e_\mu^\alpha$ about a noninvertible rank-1 matrix with only one nonzero value:

$$e_\mu^\alpha = \begin{pmatrix} 0 & 0 & 0 & 0 \\ 0 & 0 & 0 & 0 \\ 0 & 0 & 0 & 0 \\ 0 & 0 & 0 & \lambda \end{pmatrix}_\mu^\alpha + A_\mu^\alpha = \delta_3^\alpha \bar{e}_\mu + A_\mu^\alpha, \quad \text{where} \quad \bar{e}_\mu = \lambda \delta_\mu^3. \qquad (80)$$

$A_\mu^\alpha$ will be thought of as a small perturbation. This is an expansion about a singular spacetime geometry; Minkowski space is described by an identity matrix $e_\mu^\alpha = \delta_\mu^\alpha$. Next we expand the frame field about zero:

$$\omega_\mu^{\alpha\beta} = \begin{cases} B_\mu^0 & \alpha, \beta = 1, 2 \\ B_\mu^1 & \alpha, \beta = 0, 2 \\ B_\mu^2 & \alpha, \beta = 0, 1 \end{cases}. \qquad (81)$$

We are essentially just relabelling the $\omega_\mu^{\alpha\beta}$ fields in terms of $B_\mu^k$ fields.

If we linearly expand the Tetradic Palatini action in this way, and only keep terms that are quadratic in $A$ and $B$, then we obtain the following foliated Lagrangian:

$$L = \sum_{k=0,1,2} \epsilon^{\mu\nu\rho\sigma} \bar{e}_\mu B_\nu^k \partial_\rho A_\sigma^k, \qquad (82)$$

where $\bar{e}_\mu$ was defined in Eq. (80). This is very similar to the first term in the foliated field theory [Eq. (3)] for a single foliation, which roughly corresponds to a single stack of toric codes on a lattice.

This suggests that in this singular limit [Eq. (80)], tetradic Palatini gravity has a gapped energy spectrum (with no gravitons) and exhibits a ground state degeneracy that is exponential large with the length of the system. The possible existence of this large amount of degeneracy may not be surprising since gravity and linearized gravity have recently been argued to exhibit an extensive amount of ground state degeneracy [89–91].

# D   Table of Hamiltonian Terms

For reference use, we show all of the string-membrane-net model operators together:

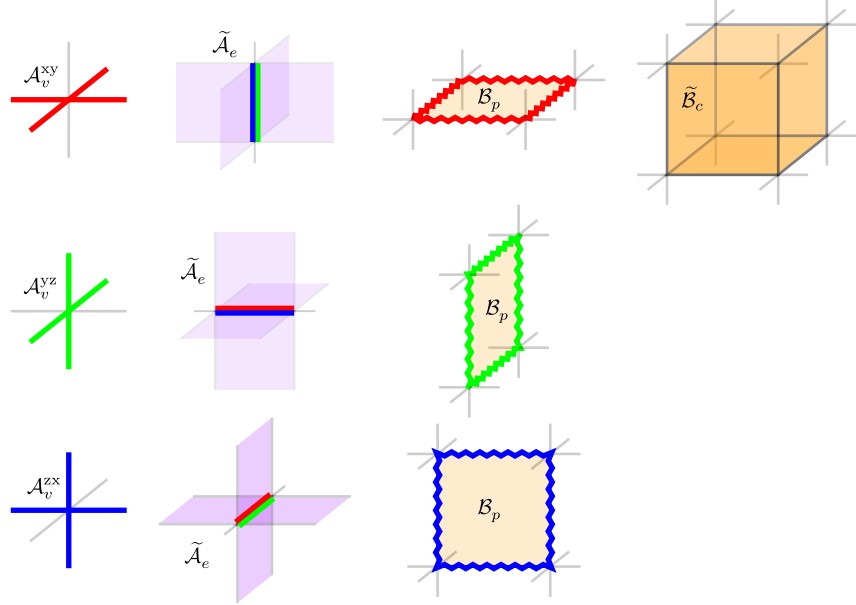

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
