# Peer review of "Foliated Field Theory and String-Membrane-Net Condensation Picture of Fracton Order"

_SciPost Physics, doi:SciPost Phys. 6, 043 (2019)_

## Round 2 · Referee Report · Anonymous (Referee 1) · 2019-2-22

Strengths

1- Introduces two novel and potentially useful formalisms 2- Raises several interesting questions which the new formalisms seem well-equipped to answer 3- Very well-written

Weaknesses

1- Does not actually show (only suggests) that the new formalisms are strong enough to produce results beyond those of other frameworks 2- Motivation of gauge symmetry structure in the foliated field theory is weak (see report)

Report

This paper introduces two interrelated new formalisms in the context of fractons: foliated field theories and string-membrane-net condensates. The two are used in tandem to provide a novel perspective on the X-Cube model. The new formalisms are interesting and suggest a number of interesting questions and generalizations, some of which (e.g. dynamical foliations) are difficult to imagine attacking using previously available frameworks. The X-Cube model has been studied exhaustively, so this work places well-understood phenomena into novel frameworks rather than creating a new model or predicting new phenomenology. As such, I think that this paper is valuable to the community as written, though it would be more impactful if the authors could show that their framework is indeed powerful enough to answer one of the questions they raise or produce a new generalization.

The paper is very well-written. The connection between the field theory and the string-membrane-net picture is well-motivated, and each of the two major sections flows logically and is clear in its exposition. One major conceptual point, though, that I found lacking was the motivation of the gauge symmetries in Eq. (6). Indeed one can simply obtain these gauge symmetries by inspection. However, it is quite unusual that transforming A and b under their standard gauge transformations requires that a and B (respectively) also transform. To what extent, then, should I think of these as independent gauge fields? What is the intuition for why the gauge transformation rules should couple in this way? Is it important that it is precisely when b transforms nontrivially that the corresponding transformation on A cannot be absorbed into a conventional gauge transformation on A?

Requested changes

1- Add discussion of the physics of the gauge transformation rules in Eq. (6) (see report)

2- It would be helpful in both the field theory section and the (dual) string-net section to add small comments about how one can see that the fractons should be created in sets of four.

3- In section 4.1.1, the authors suggest that the ground state degeneracy might be calculated without imposing a cutoff. I do not understand how this could make sense because the degeneracy, a dimensionless number, depends on the system size, which is only dimensionless in the presence of a cutoff.

---

## Round 2 · Referee Report · Anonymous (Referee 2) · 2019-3-11

Strengths

1- The authors introduce a new and interesting kind of field theory, which they demonstrate to describe foliated (type I) fracton states. The field theory elegantly takes advantage of the foliation information to couple together familiar topological field theories.

2- The authors then make contact between these field theories and a class of solvable lattice models, which they show to be equivalent to previous constructions of type I fracton states.

3- The construction suggests many generalizations.

Weaknesses

1- The paper does not describe any previously-undiscovered phenomena or models or states, and does not actually pursue the many generalizations its formalism suggests.

Report

This is an interesting paper which proposes a new continuum field theory description of type I (foliated) fracton models.
The new ingredient in the field theory description is a collection of static background fields $e^k_\mu$ which
encode a foliation structure.

The authors do a nice job of explaining the relation between this very simple and appealing construction and a class of solvable lattice models,
which they then show to be equivalent to previous constructions of such type I fracton states.

Although the paper does not describe any previously-undiscovered phenomena or models or states, it does offer a useful unification of many known constructions, which suggests possible generalizations. The way in which the mobility restrictions are realized is quite cute.
The gauge structure of the theory is interesting, if mysterious.
Relative to previous proposed fracton field theories (some by some of the same authors) the present construction clarifies the connection to more familiar topological field theories, and it clarifies the role of the extra (foliation) structure required to define these states.

The authors very briefly discuss various generalizations of their construction. I was surprised that this list did not emphasize
the possibility of adding more general Chern-Simons terms within the leaves (though this possibility does appear via the matrix $M$ in equation (38)). When there is chiral topological order in the layers of the foliation it is in general difficult to write solvable models, and the field theory description may therefore offer new vistas. (The $p+ip$ layers in reference 68 are a special case and the only example I've seen so far.)

Some more miscellaneous minor comments:

-- I don't know what to make of the discussion about realizing this foliated field theory by expanding around a singular configuration of Palatini gravity, but I guess it is safely hidden in the appendix.

-- at the beginning of section 3.2, the authors speak of "p-strings" without defining what they are.

-- the phrase "$Z_N$ qubit" is awkward. I think the common term is "qudit".

-- A sign error: I think the "+" in "equal+weight" was meant to be a "-".

Requested changes

1- I don't think that the issue of needing a regulator to compute the ground state degeneracy is related to questions of large gauge transformations. The authors should consider removing this remark.

2- Regarding the comment about "soft modes" at the end of appendix C: the groundstate degeneracy associated to this foliated field theory seems distinct from the states discussed in refs 90-92, for example because here the degenerate states are locally indistinguishable, i.e. there is topological order. I believe it is not the case that the states swept out by the asymptotic symmetries in QED have this property. I say this because soft photons are ordinary particle excitations which can be detected locally (a small detector has a finite probability of detecting a long-wavelength photon). The authors should consider removing this remark.

---

## Round 3 · Author Response

Dear Editor and Referees,

Thank you for considering and carefully reviewing our work and for the helpful comments and suggestions. Below are detailed responses to the referee reports.

Best regards,
Kevin Slagle
David Aasen
Dominic Williamson
* * *
Response to both Referees:
* * *
-- Weakness 1 from both Referees:

Both referees noted that our work does not use our formalism to study physics beyond what has already been currently studied. (Although we do not emphasize it, Appendix A actually does include new models.) Indeed, this was a deliberate choice. We think our current work already introduces a lot of new ideas with a focused theme and that including further generalizations would obfuscate the message of this work, along with lengthening the manuscript and delaying publication. We are currently working on exploring generalizations in a forthcoming work.
* * *
Response to Referee 1:
* * *
-- Weakness 2, Report, and Requested change 1:
Indeed, the gauge transformation is rather exotic. We now make note of this and point out that it is due to the third term in Eq. 3, which couples the b and A fields together. We also added footnote 6 to make more connections.

We prefer to think of A^k and a (and B^k and b) as separate gauge fields, but either viewpoint may be fine. Even in nonabelian BF theory where we treat A as a nonabelian gauge field, when A transforms we must also transform B; but we do not interpret A and B as the same gauge field.

That is an interesting observation that trivial (i.e. closed lambda) gauge transformations on b can be absorbed into the chi guage transformation on B. The consequence of this fact is that these trivial gauge transformations do not impose any additional mobility constraints on the currents beyond the constraint implied by the chi transformation. This can be seen by taking the divergence (\partial_mu) of both sides of the i^{nu,mu} current constraint (middle equality in Eq. 8) and noting that the left-hand side is zero due to the antisymmetry of i^{nu,mu} and the right-hand side is zero due to the constraint imposed by the chi transformation (left equality in Eq. 8).

-- Requested change 2:
We added a paragraph at the end of Section 2.1.1 and a sentence to Section 3.4.1 to discuss the creation of four fractons.

-- Requested change 3:
We note in Section 4.1.1 that the line integral of e^k is a dimensionless number (when e^k is dimensionless, which we think is the only sensible choice). If one considers the line integral along a non-contractable loop, then it is tempting to interpret that as an integer number of layers along a periodic direction. As such, it is feasible that imposing a cutoff may not be necessary. We modified Section 4.1.1 to try to make this more clear.
* * *
Response to Referee 2:
* * *
-- Report:
We only mentioned p-strings to note a connection to previous work. We moved this comment to a footnote. Thank you for mentioning the Z_N qubit and "equal+weight" typos; we have fixed them.

-- Requested change 1:
We removed the comment of large gauge transformations since it is indeed rather speculative.

-- Requested change 2:
We removed the mention of soft modes since we're only trying to make a connection to the large ground state degeneracy in linearized gravity that was argued to exist in Ref 90.

Although a finite-energy photon has a finite probability of being detected by a finite-sized detector during a finite amount of time, this probability approaches zero as the energy of the photon approaches zero. Thus, low-energy photon states can be arbitrarily locally indistinguishable in the low-energy limit. (Ref 90 also argues that U(1) gauge theory has a large ground state degeneracy.)

-- Communication via editor:
The referee asked the following question via a communication with the editor:
"In their lagrangian (3), why do they not consider an additional coupling of the form ∑kek∧b∧Bk ? I suppose that, if its coefficient is properly quantized, it can be removed by a rotation between Ak and Bk, but this acts on the charge lattice of the theory on the leaves. I think this merits some comment."

This kind of term is briefly mentioned in Eq. 38 as a possible future direction. Simply adding it to our Lagrangian (Eq. 3) could drastically change the theory since it is not invariant under the lambda gauge transformation in Eq. 6. We did not consider it for this reason and because it was not motivated by the scope of this work.

---

## Round 3 · List of Changes

1) We added a summary of the string-membrane-net model in the introduction.

2) We elaborate on the gauge transformation after Eq. 6.

3) We reordered the equalities in Eq. 6-8 and 12 to make the ordering more intuitively simple.

4) We added discussion of the 4-fold fracton creation at the end of Section 2.1.1 and 3.4.1.

5) We moved the p-string discussion at the beginning of Section 3.2 to a footnote.

6) We clarified the cutoff/quantization issues in 4.1.1.

7) We defined k in Eq. 76 more explicitly.

8) We removed discussion of soft modes at the end of Appendix C.

A detailed markup of the changes can be found at:
https://drive.google.com/file/d/13HtnUac8-dna3EG8CTwv4Kr3nu961QpC

Resubmission 1812.01613v3 on 21 March 2019
Submission 1812.01613v2 on 24 January 2019

---

## Editorial Decision

published